# Combining genomics and epidemiology to analyse bi-directional transmission of *Mycobacterium bovis* in a multi-host system

Joseph Crispell[1], Clare H Benton[2], Daniel Balaz[3], Nicola De Maio[4], Assel Ahkmetova[5], Adrian Allen[6], Roman Biek[5], Eleanor L Presho[6], James Dale[7], Glyn Hewinson[8], Samantha J Lycett[3], Javier Nunez-Garcia[9], Robin A Skuce[6], Hannah Trewby[10], Daniel J Wilson[11], Ruth N Zadoks[5], Richard J Delahay[2]*, Rowland Raymond Kao[3,12]*

[1]School of Veterinary Medicine, Veterinary Sciences Centre, University College Dublin, Dublin, Ireland; [2]National Wildlife Management Centre, Animal & Plant Health Agency (APHA), London, United Kingdom; [3]Roslin Institute, University of Edinburgh, Edinburgh, United Kingdom; [4]European Molecular Biology Laboratory, European Bioinformatics Institute (EMBL-EBI), Cambridge, United Kingdom; [5]Institute of Biodiversity, Animal Health & Comparative Medicine, College of Medical, Veterinary & Life Sciences, University of Glasgow, Glasgow, United Kingdom; [6]Agri-Food & Biosciences Institute Northern Ireland (AFBNI), Belfast, United Kingdom; [7]Animal & Plant Health Agency (APHA), London, United Kingdom; [8]Centre for Bovine Tuberculosis, Institute of Biological, Environmental and Rural Sciences, University of Aberystwyth, Aberystwyth, United Kingdom; [9]Genomics Medicine Ireland, Dublin, Ireland; [10]Quadram Institute Bioscience, Norwich, United Kingdom; [11]Big Data Institute, Li Ka Shing Centre for Health Information and Discovery, Nuffield Department of Population Health, University of Oxford, Oxford, United Kingdom; [12]Royal (Dick) School of Veterinary Studies, University of Edinburgh, Edinburgh, United Kingdom

*For correspondence:
Dez.Delahay@apha.gsi.gov.uk (RJD);
rowland.kao@ed.ac.uk (RRK)

**Competing interests:** The authors declare that no competing interests exist.

**Abstract** Quantifying pathogen transmission in multi-host systems is difficult, as exemplified in bovine tuberculosis (bTB) systems, but is crucial for control. The agent of bTB, *Mycobacterium bovis*, persists in cattle populations worldwide, often where potential wildlife reservoirs exist. However, the relative contribution of different host species to bTB persistence is generally unknown. In Britain, the role of badgers in infection persistence in cattle is highly contentious, despite decades of research and control efforts. We applied Bayesian phylogenetic and machine-learning approaches to bacterial genome data to quantify the roles of badgers and cattle in *M. bovis* infection dynamics in the presence of data biases. Our results suggest that transmission occurs more frequently from badgers to cattle than *vice versa* (10.4x in the most likely model) and that within-species transmission occurs at higher rates than between-species transmission for both. If representative, our results suggest that control operations should target both cattle and badgers.

**eLife digest** Disease-causing microbes that infect more than one type of animal can be difficult to control. This is especially true when they infect wildlife. For example, *Mycobacterium bovis* is a bacterium that causes tuberculosis in tens of thousands of cattle in Britain every year and also infects badgers and other wildlife. Controlling the infections in cattle is essential, as it helps prevent the bacteria from infecting humans, improves cattle welfare and reduces the substantial costs to the livestock industry.

Analysing the relatedness of *M. bovis* genomes from infected cattle and badgers may help scientists work out how often badgers infect cattle and vice versa. Scientists have collected data and *M. bovis* samples from infected badgers in Woodchester Park, in England, for over three decades. Using these data and additional information about *M. bovis* infecting nearby cattle may help scientists learn how the bacteria spreads and how to stop it.

Now, Crispell et al. show that complex patterns of contact between cattle and badgers likely drive the persistence of tuberculosis in cattle, also known as bovine tuberculosis. In three separate analyses, Crispell et al. compared the genomes of *M. bovis* found in cattle and badgers, the animals' locations, when they were infected, and whether they could have been in contact. The analyses found that *M. bovis* was likely to have been transmitted more frequently from badgers to cattle rather than from cattle to badgers. They also showed that transmission within each species happened more often than transmission between species.

If these results are confirmed by other studies, they may help scientists develop better strategies for controlling tuberculosis in British cattle. In particular, controversial control strategies – such as badger culls – could be more targeted to better combat tuberculosis in cattle but have less of an impact on badgers. These insights might also aid control efforts in other countries where bovine tuberculosis is a problem and an important source of human tuberculosis.

## Introduction

Control of a pathogen in a system where it can infect multiple species requires an understanding of the role of each host species in the infection dynamics (*Haydon et al., 2002*). For example, when each host species is capable of maintaining infection independently, control operations in one species can be rendered ineffective as a result of spillover from another. *Mycobacterium bovis* infection in cattle populations (resulting in bovine tuberculosis - bTB) is a problem around the world (*Ayele et al., 2004*; *Cousins and Roberts, 2001*; *de Kantor and Ritacco, 2006*; *Godfray et al., 2013*; *Reviriego Gordejo and Vermeersch, 2006*; *Schmitt et al., 2002*), with many wildlife species implicated in its spread and persistence in different bTB systems (*Delahay et al., 2002*; *Gortazar et al., 2003*; *Miller and Sweeney, 2013*; *Nugent, 2005*; *Nugent et al., 2015*). On the islands of Britain and Ireland, the current evidence suggests that effective control of infection in cattle is hindered by transmission from an infected wildlife population – the European badger (*Meles meles*) (*Godfray et al., 2013*).

Although a considerable amount of research demonstrates an association between *M. bovis* found in sympatric cattle and badger populations (*Balseiro et al., 2013*; *Goodchild et al., 2012*; *Olea-Popelka et al., 2005*; *Vial et al., 2011*; *Woodroffe et al., 2005*), quantification of the direction and extent of transmission remains elusive. Recent studies using whole genome sequences (WGS) have demonstrated a close genetic relationship among *M. bovis* isolates taken from sympatric cattle and wildlife populations (*Biek et al., 2012*; *Glaser et al., 2016*; *Patané et al., 2017*). However, the low genomic variability of *M. bovis* and imbalanced sampling across host species has limited the ability to identify the direction of transmission. Evidence to date suggests that, even with access to pathogen sequence data, obtaining directional estimates of transmission might only be possible at the population level and will require dense targeted sampling and fine-grained epidemiological metadata (*Kao et al., 2016*; *Kao et al., 2014*), as has previously been demonstrated in investigations of *M. tuberculosis* outbreaks in humans (*Bryant et al., 2013*; *Gardy et al., 2011*; *Guthrie et al., 2018*; *Walker et al., 2012*; *Walker et al., 2018*; *Yang et al., 2017*) and in tracing between cattle herds for outbreaks of *M. bovis* (*Biek et al., 2012*; *Salvador et al., 2019*). However,

these approaches have yet to be applied to situations where dense multi-host pathogen data are available.

Since the 1970s, a high-density naturally infected badger population at Woodchester Park in southwest England has been the subject of detailed study (*Delahay et al., 2013*). Both the resident badgers and sympatric cattle herds are frequently infected with *M. bovis,* providing the potential for inter-species transmission of infection to occur in either direction (*DEFRA, 2017*; *Delahay et al., 2013*). The data and samples associated with bTB occurrence in and around Woodchester Park are uniquely detailed, with individual-level host life history data and archived *M. bovis* isolates available for both the cattle (*Orton et al., 2018*) and badger (*Delahay et al., 2013*) populations. By combining WGS of selected cattle and badger isolates, with detailed local population data from this exceptionally in-depth study system, our work aimed to quantify the relative roles of the local badger and cattle populations in the spread and persistence of *M. bovis* in an endemic area.

Based on previous evidence of transmission between cattle and badgers, and the success of combining detailed tracing methods with WGS for *M. tuberculosis*, our hypothesis is that *M. bovis* circulation in our endemic setting is not limited to a single maintenance host and that it involves bidirectional transmission between the two host populations. Our research aimed to test this hypothesis and to quantify transmission patterns by analysing the Woodchester Park data using a series of statistical and observational analyses linking pathogen genome data with diagnostic testing and population movement and demographic data for both cattle and badgers.

## Results

### Selecting the isolates, generating and processing the sequencing data

Archived *M. bovis* isolates were available from 116 badgers and 189 cattle living in and around Woodchester Park. Multiple isolates were available from the sampled badgers, resulting in a total of 230 isolates sourced from badgers. These isolates were whole genome sequenced, and, after quality assessments, 193 badger-derived (from 98 individual badgers taken from 2000 to 2011) and 159 cattle-derived sequences (from 1988 to 2013) were retained for further analyses.

### Evidence of epidemiological signatures in the genetic data

To investigate the presence of spatial, temporal, and network signatures associated with infection dynamics in the *M. bovis* genomic data, inter-sequence genetic distances were calculated between all the cattle- and badger-derived sequences and compared to population metrics. The metrics described the spatial-, temporal-, and network-based relationships that were expected to be associated with pathogen transmission. The genetic and epidemiological data were compared using Random Forest (*Liaw and Wiener, 2002*) and Boosted Regression (*Elith et al., 2008*) models in R (v3.4.3; *R Development Core Team, 2016*) to separately analyse badger–badger (n = 12483), cattle–cattle (n = 1927), and badger–cattle (n = 4838) comparisons.

The Random Forest (and Boosted Regression) models were able to explain approximately 67% (62%), 60% (54%) and 75% (70%) of the variation observed in the inter-sequence genetic distance distributions associated with the badger–badger, cattle–cattle, and badger–cattle comparisons, respectively. For each of these models, metrics based on spatial and temporal distances were the most informative in explaining the variation in the genetic distances. Generally, as the temporal and spatial distances associated with the sampled animals decreased, the number of differences between the *M. bovis* genomes decreased (*Appendix 1—figures 5*, *6* and *7*). There was substantial agreement in the variable rankings between the Random Forest and Boosted Regression models (*Appendix 1—figures 2*, *3* and *4*). For the within-species comparisons metrics, the network data were also highly informative. Generally, the number of differences between the genomes associated with a pair of animals of the same species decreased as the connectedness of their social groups (badgers) or herds (cattle) increased. The variation explained by the Random Forest models and the high ranking of spatial-, temporal-, and network-based metrics was robust to the presence of highly correlated or non-informative metrics and those with missing data (data not shown).

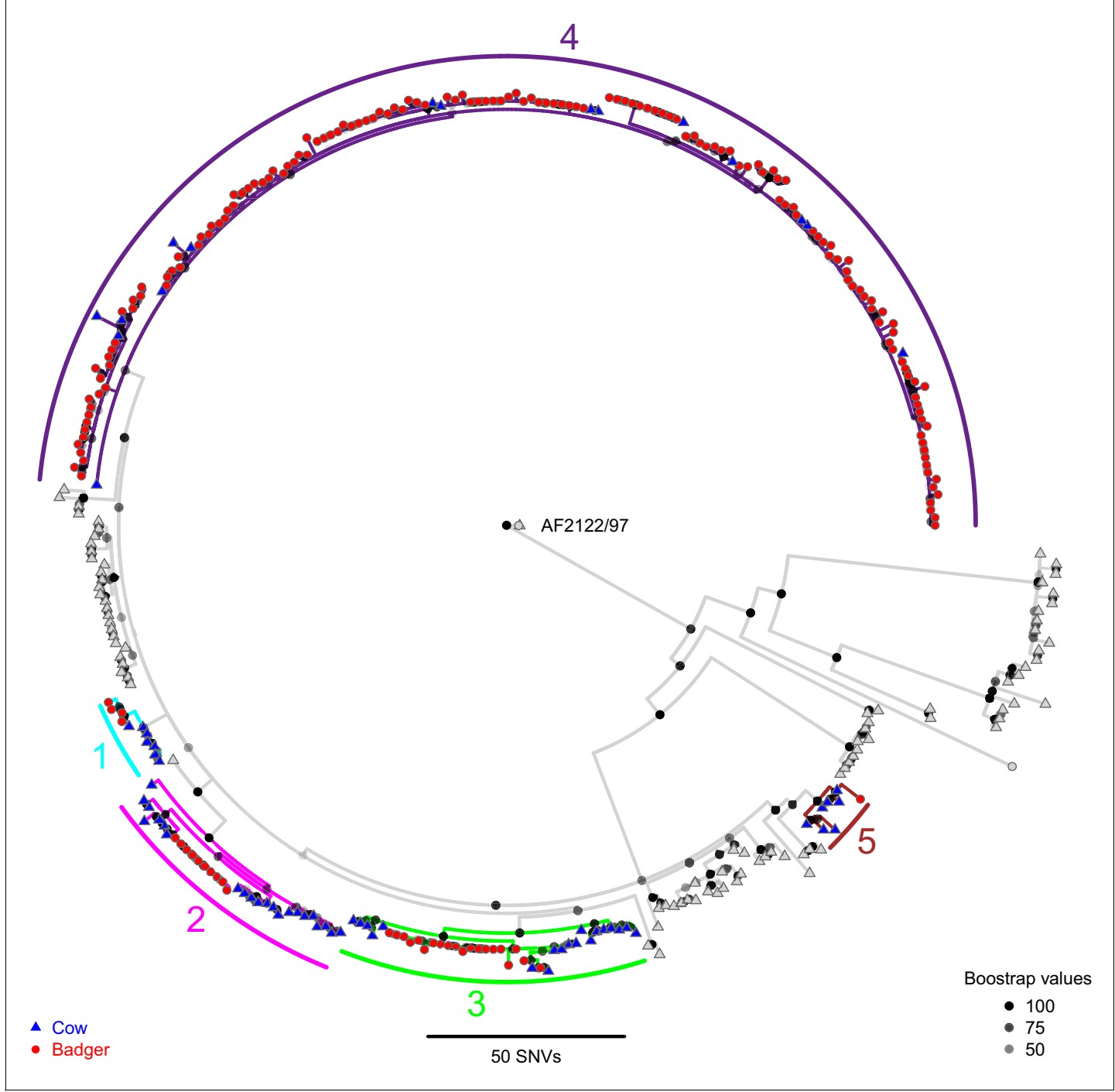

**Figure 1.** A Maximum Likelihood phylogenetic tree constructed using RAxML (v8.2.11; *Stamatakis, 2014*) and rooted against the *Mycobacterium bovis* reference sequence, AF2122/97 (*Malone et al., 2017*). Badger and cattle isolates are represented at the tips of the phylogeny by circles and triangles, respectively. Five clades, labelled 1–5, are highlighted with cyan, pink, green, purple, and brown branches, respectively. Cattle and badger isolates within the clades can be distinguished by their shape and colour. Each internal node in the phylogeny is shown as a grey to black shaded circle, with the intensity of the shading indicating the amount of support each node had across 100 bootstraps.

The online version of this article includes the following figure supplement(s) for figure 1:

**Figure supplement 1.** Each of the clades from *Figure 1* in the main manuscript are plotted separately.

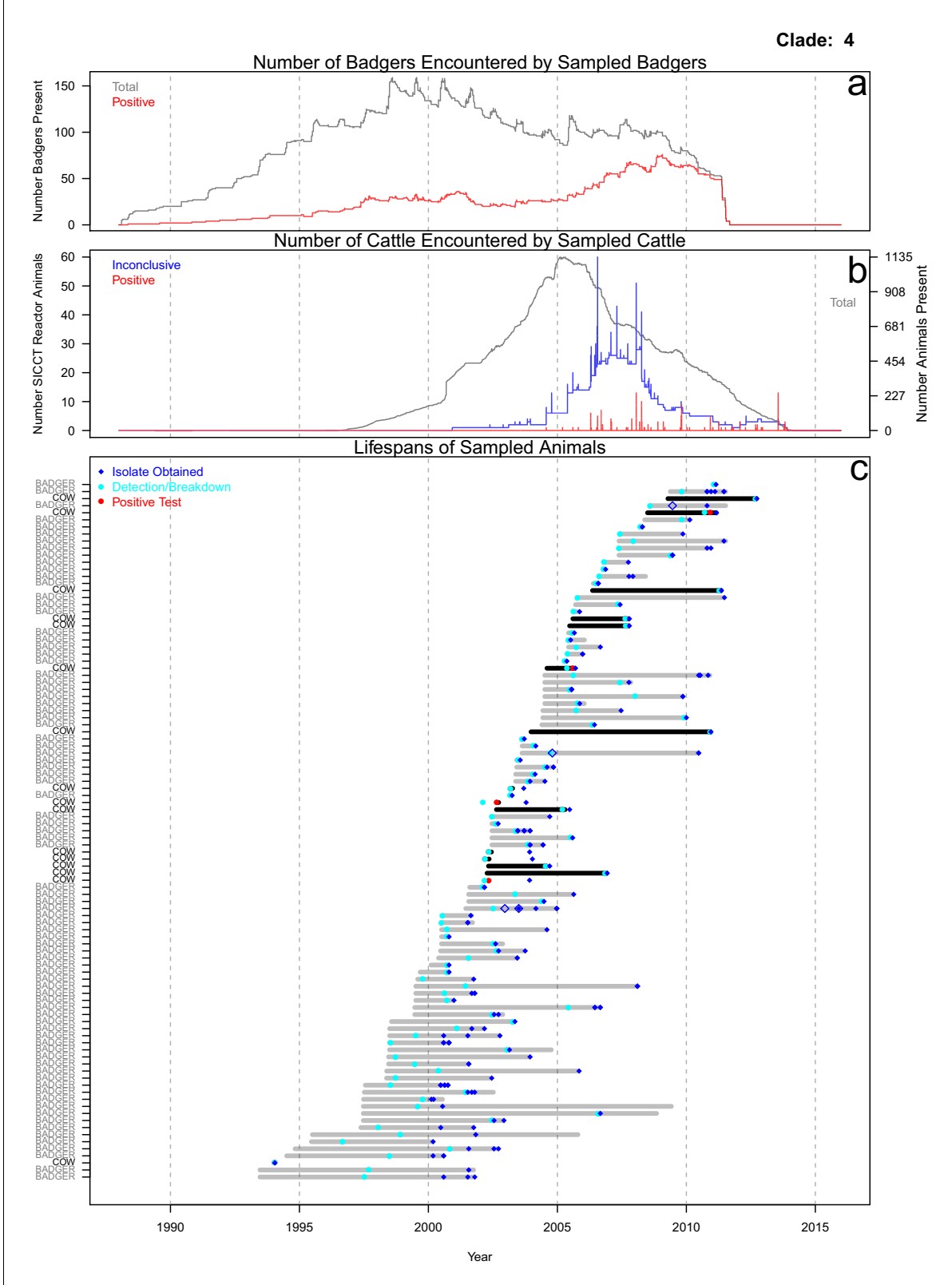

**Figure 2.** Life history summaries of the sampled and in-contact cattle and badgers associated with clade 4 in *Figure 1*. (a) The number of in-contact badgers associated with the sampled badgers (total in grey, number of animals that have tested positive in red). (b) The number of in-contact cattle associated with the sampled cattle (total in grey [right axis], number of animals that reacted inconclusively [red] or positively [blue] to routine skin test
*Figure 2 continued on next page*

*Figure 2 continued*

[left axis]). In-contact animals are those that lived in the same herd (cattle) or social group (badgers) at the same time as the sampled animals. (c) The recorded lifespans of the sampled cattle (black horizontal bars) and badgers (grey horizontal bars) associated with clade 4.

The online version of this article includes the following figure supplement(s) for figure 2:

**Figure supplement 1.** Life history summaries of the sampled and in-contact cattle and badgers associated with clade 1 in *Figure 1*.
**Figure supplement 2.** Life history summaries of the sampled and in-contact cattle and badgers associated with clade 2 in *Figure 1*.
**Figure supplement 3.** Life history summaries of the sampled and in-contact cattle and badgers associated with clade 3 in *Figure 1*.
**Figure supplement 4.** Life history summaries of the sampled and in-contact cattle and badgers associated with clade 5 in *Figure 1*.

## Inter-species clades identified in the phylogeny

The relatedness of *M. bovis* genomes sampled from the cattle and badgers was evaluated by constructing a phylogenetic tree (*Figure 1*) using RAxML (v8.2.11; *Stamatakis, 2014*). Genetic diversity was observed between the cattle- and badger-derived *M. bovis* sequences, with the number of Single Nucleotide Variants (SNVs) between sequences ranging from 0 to 150 (median = 20). Five clades including cattle- and badger-derived sequences were identified (*Figure 1* and *Figure 1—figure supplement 1*), using a 10 SNV threshold (informed by thresholds used for *M. tuberculosis* [*Bryant et al., 2013*; *Jajou et al., 2018*; *Roetzer et al., 2013*; *Yang et al., 2017*]).

Four of the five clades (1–4) identified contained highly similar (within three SNVs) badger- and cattle-derived *M. bovis* sequences. The badger-derived *M. bovis* sequence in clade 5 was six SNVs away from its closest cattle-derived sequence. The similarities between the cattle-derived and badger-derived *M. bovis* sequences in clades 1–4 indicate recent shared transmission histories (*Meehan et al., 2018*). Clade 4 (highlighted in purple in *Figure 1*) contained the majority (156/193) of the badger-derived *M. bovis* sequences and represents the main lineage circulating within the Woodchester Park badger population. In addition, the presence of 16 cattle-derived sequences in clade 4, 15 of which were distant (up to 12 SNVs) from the clade root is consistent with multiple badger-to-cattle transmission events. In contrast, the presence of cattle-derived sequences close to the roots of clades 1–5 suggests that these lineages might have originated in cattle, although these patterns could also be explained by the cattle population being sampled up to 12 years prior to the badger population (cattle were sampled from 1988 to 2013 and badgers from 2000 to 2011). Although clades 1 and 5 contained highly similar sequences originating from cattle and badgers, each clade was associated with only eight animals, making meaningful inference of inter-species transmission patterns difficult. In addition to inter-species clades, several cattle-only clades were identified (*Figure 1*).

Consistent with our hypothesis, the close proximity of *M. bovis* genomes sourced from cattle and badgers suggests that inter-species transmission occurred in the sampled system. In addition, the presence of clades dominated by a single species suggests that sustained within-species transmission has been occurring in both the cattle and badger populations.

The life histories of the sampled cattle and badgers and in-contact animals associated with the inter-species clades (clades 1–5) identified in *Figure 1* were interrogated. In this manuscript, a badger or cow is considered 'sampled', if one of the *M. bovis* genomes analysed here was sourced from it. In-contact animals were defined as those that lived in the same herd (for cattle) or social group (for badgers) at the same time as one or more of the sampled animals, according to the available data. From the interrogations of the life history data, further evidence indicative of inter-species transmission and disease maintenance in the Woodchester Park badger population was identified for the animals associated with clade 4 (*Figure 2*; equivalent figures for the remaining clades can be found in *Figure 2—figure supplements 1*, *2*, *3,* and *4*). Infection was detected in the majority of the sampled badgers before it was detected in the majority of the sampled cattle. Sampled badgers were present in Woodchester Park at least from 1993 until 2011, based on the available capture and sampling data (*Figure 2c*). The sampled badgers were in contact with 575 captured badgers, 291 (51%) of which had tested positive for *M. bovis* infection at some point in their lives (*Figure 2a*). In contrast, the sampled cattle were in contact with 1760 cattle, of which only 312 (18%) tested positive for *M. bovis* (*Figure 2b*). In the animals associated with clade 4, infection was detected earlier in badgers, except in the case of one cow, despite the cattle population being sampled over a broader temporal and spatial window (see Materials and methods section: 'Selecting the isolates' for more

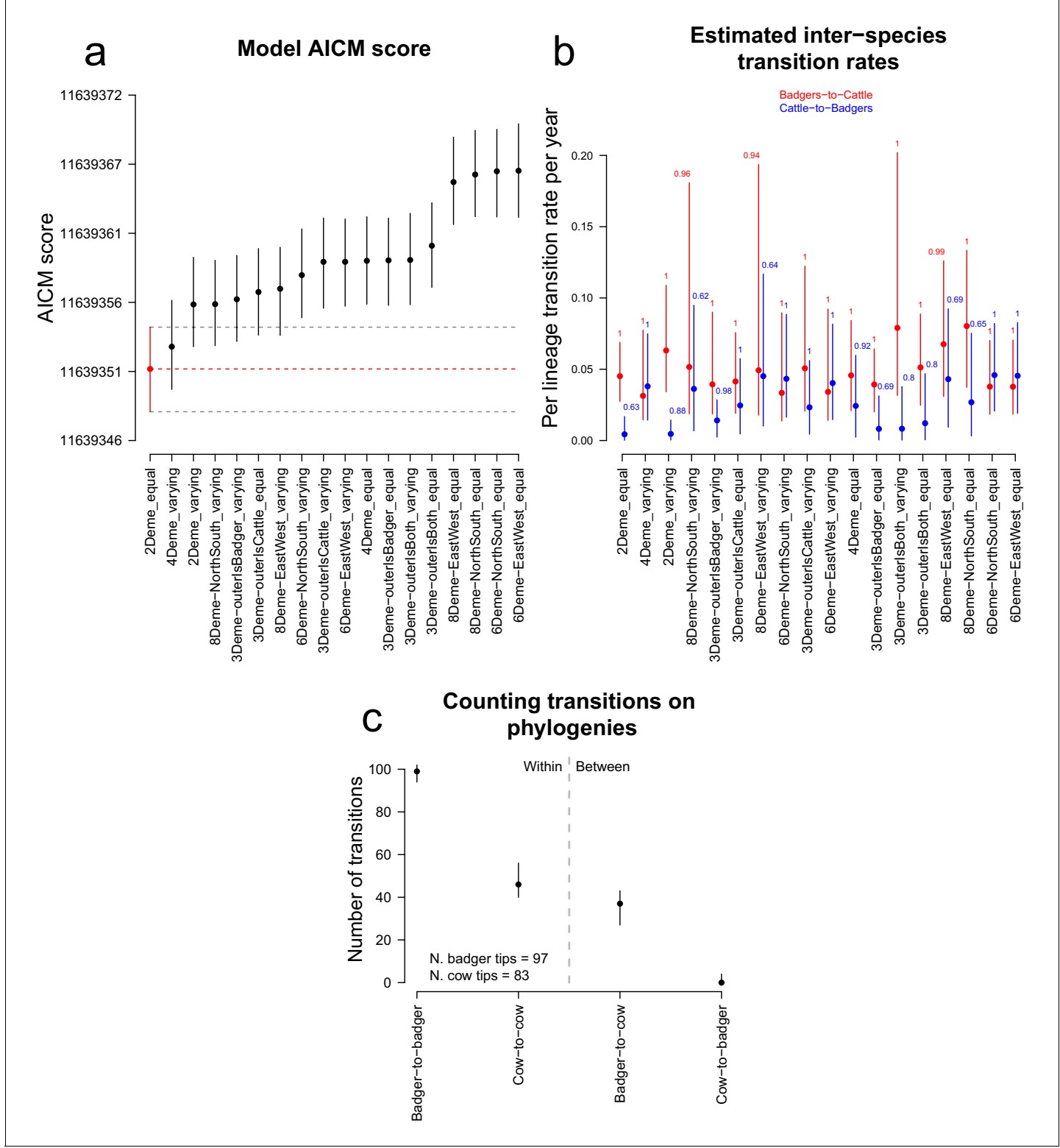

**Figure 3.** Comparison of likelihood scores and inter-species transition rate estimates from the BASTA analyses. Model structure is described in *Figure 6*, and for each model the sizes of defined demes were held equal or allowed to vary. (a) The Akaike Information Criterion Markov Chain Monte Carlo (AICM; *Baele et al., 2013*) scores (lower is better) calculated for each of the representations of a structured population analysed in BASTA (*Figure 6*). The vertical lines show the lower and upper (2.5% and 97.5%, respectively) bounds of the AICM scores computed on 100 bootstrapped posterior likelihoods. (b) Estimated inter-species transition rates for each model. Where multiple badgers-to-cattle and cattle-to-badgers transition rates

*Figure 3 continued*

were estimated (see *Figure 6*), the values were summed. The values above each vertical line represent the posterior probability of each rate, either as a mean of probabilities associated with multiple estimated rates (for the 3Deme_outerIsBadgers, 4Deme, 6Deme, and 8Deme models) or a single probability (for the 2Deme, 3Deme_outerIsBoth, and 3Deme_outerIsCattle models). (c) The number of transitions between the known and estimated states counted on each phylogenetic tree in the posterior distribution produced by the '2Deme_equal' structured population model analysed in BASTA (counting is illustrated in *Figure 3—figure supplement 1*). The vertical lines show the lower and upper (2.5% and 97.5%, respectively) bounds of the distributions.

The online version of this article includes the following figure supplement(s) for figure 3:

**Figure supplement 1.** Diagrams illustrating how the transmission events were counted on each of the phylogenies in the posterior distributions produced by BASTA.

details). In addition, the badgers were the most represented species in clade 4. These two observations suggest that the clade 4 lineage was being maintained in the badger population. The single cattle-derived sequence that was found closest to the root node of clade 4 (*Figure 2c*) was sourced from an animal sampled six years prior to any sequences derived from badgers being available. Across all inter-species clades investigated, the sampled cattle (n = 71) were in contact with approximately 11,732 animals, 1356 of which tested positive for *M. bovis* infection, whereas the sampled badgers (n = 97) were in contact with approximately 650 badgers, over half of which (329) tested positive.

## Estimated inter-species transmission rates

Although the patterns observed in the phylogenetic and animal life history data were consistent with inter-species transmission in both directions, further analyses were required to quantify the inter-species transmission rates. These further analyses should account for the temporal and spatial sampling biases resulting from the broader sampling window applied to the cattle population in time (1988 to 2013 versus 2000 to 2011) and space (cattle were sampled from up to 100 km away from the Woodchester Park area, whereas the badgers were only sampled from within Woodchester Park).

A series of analyses were conducted using the Bayesian Structured coalescent Approximation, or BASTA, package (*De Maio et al., 2018*) available as part of Bayesian evolutionary analyses platform BEAST2 (Bayesian Evolutionary Analysis by Sampling Trees; *Bouckaert et al., 2014*). These analyses aimed to estimate the *M. bovis* inter-species transmission rates between the sampled badger and cattle populations. BASTA is capable of estimating evolutionary dynamics in a structured population and accounting for sampling biases. Here the sampled *M. bovis* population was structured as it was circulating largely separately in the sampled cattle and badger populations, as seen in *Figure 1* and the strong population-specific epidemiological signatures found by the Random Forest and Boosted Regression analyses. In addition, further structure exists within the cattle and badger populations as these were subdivided into herds and social groups, respectively. A series of increasingly spatially structured population models were defined to determine whether the inter-species transmission rates estimated using BASTA were affected by the spatial patterns evident from the Random Forest and Boosted Regression analyses. Structured population models were also chosen to address the spatial sampling biases, by introducing an increasingly structured unsampled badger population. Previous analyses have used BASTA in a similar fashion to estimate evolutionary dynamics in the presence of unsampled populations (*De Maio et al., 2015*). To further reduce the influence of the spatial and temporal biases and the computational load, the BASTA analyses used a subset of the cattle- (n = 83) and badger-derived (n = 97) *M. bovis* sequences obtained between 1999 and 2014 within 10 km of Woodchester Park.

The AICM (Akaike's Information Criterion Markov Chain Monte Carlo) score (*Baele et al., 2013*) was used to compare the BASTA analyses based on different structured populations (*Figure 3a*). The structured population with two demes (*M. bovis* populations in badgers and cattle) had the best (lowest) AICM score, although there was considerable overlap with the bootstrapped AICM score interval for one of the four deme models (splitting the *M. bovis* populations in badgers and cattle into inner and outer populations based on being within or beyond 3.5 km from Woodchester Park [*Figure 3a*]). The estimated inter-species transition rates provided from each BASTA analysis demonstrated considerable variation, with some estimated cattle-to-badger transition rates bounding zero (*Figure 3b*). The estimated transition rates can be considered equivalent to the transmission rates,

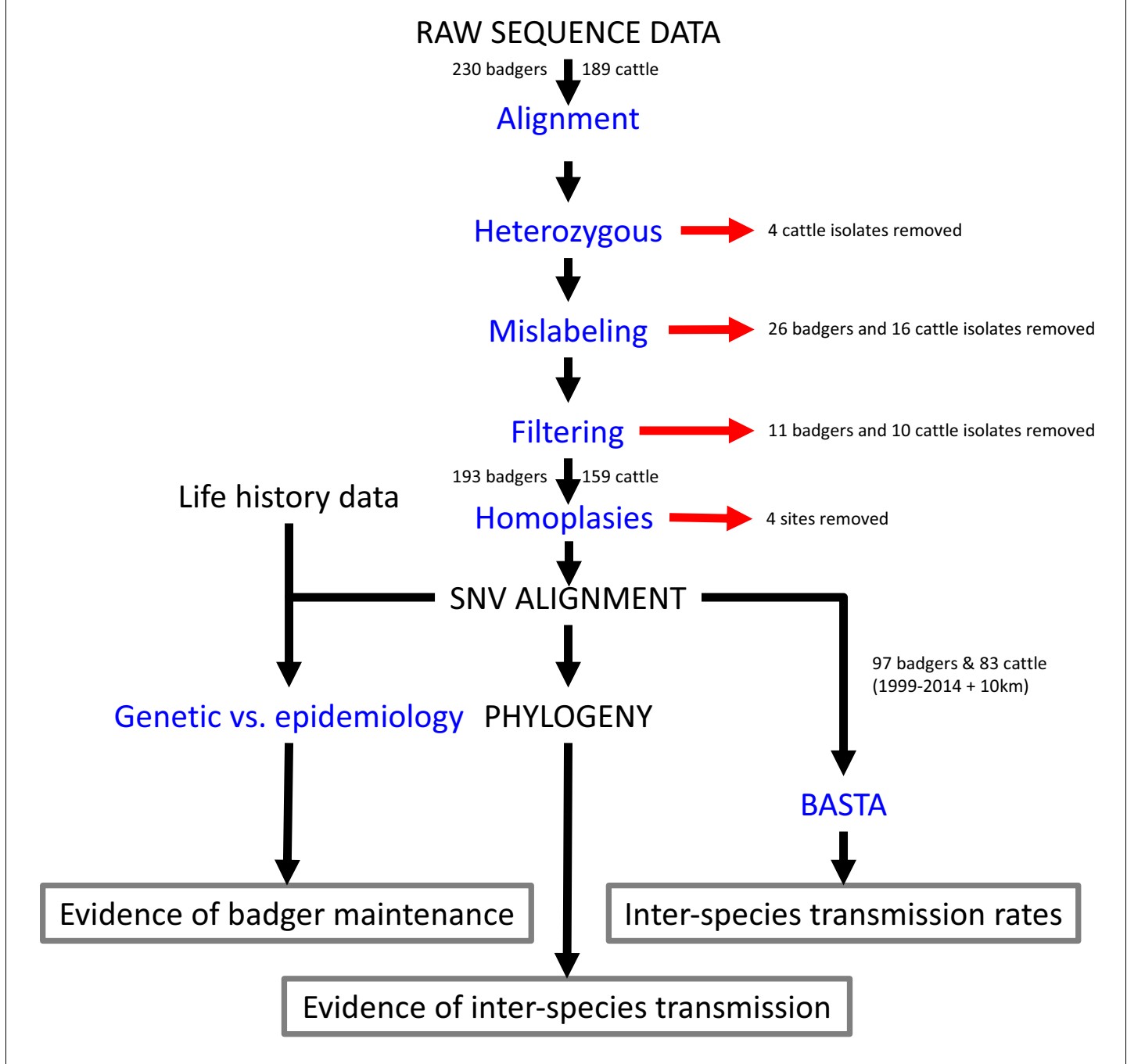

**Figure 4.** Steps involved in the analysis of *M.bovis* whole genome sequences and epidemiological data. Analyses are shown in blue and outputs and inputs in black. Red arrows represent the removal of data. The three main outputs are highlighted with grey boxes. SNV: Single Nucleotide Variant. BASTA: Bayesian Structured coalescent Approximation.

because the states (between which the transition rates were estimated) considered here represented different species. The estimates of the inter-species transition rates from the two-deme model with the best AICM score support the existence of both badger-to-cattle transmission (0.045 times per lineage per year, lower 2.5%: 0.028, upper 97.5%: 0.069) and cattle-to-badger transmission (0.0044 times per lineage per year, lower 2.5%: 0.00021, upper 97.5%: 0.017). *Figure 3b* shows the order of magnitude differences between the estimated inter-species transmission rates, with the highest supported two-deme model estimating that badger-to-cattle transmission events occurred on average

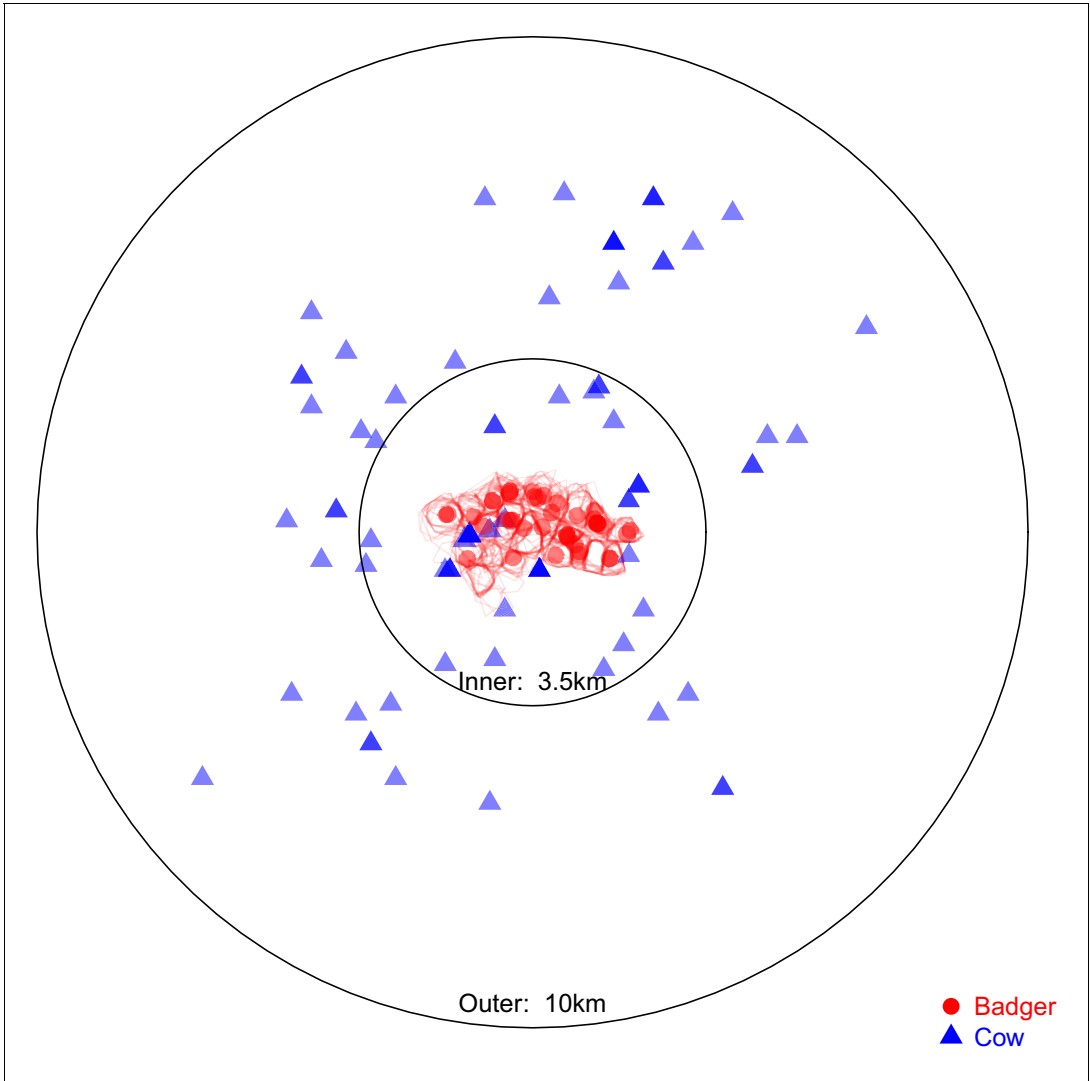

**Figure 5.** Sampling locations of the 97 badgers and 83 cattle associated with the *Mycobacterium bovis* sequences selected for analysis in BEAST2. Location represents the registered address of each sampled farm or the centroid of the estimated sampled badger social group's territory boundary (indicated by the red polygons). The overlaid circles were used to split the cattle- and badger-derived *M. bovis* sequences into 'inner' and 'outer' populations, the distances refer to the radius of each circle. The 'inner' circle was defined such that it contained all the locations associated with the available badger-derived and closest (within the badger's recorded home range of <1 km$^2$ [*Gittleman and Harvey, 1982*; *Garnett et al., 2005*; *Macdonald et al., 2008*; *Roper et al., 2003*]) surrounding cattle-derived *M. bovis* sequences.

10.4 times more frequently than cattle-to-badger transmission events in the sample population. *Figure 3c* represents the lower bound on the number of times (according to the analyses based on the favoured two-deme model) that the sampled *M. bovis* population was transmitted from one animal to another (regardless of sub-population and, where possible, assuming the ancestral node and one of its daughter nodes represent infection in the same animal [*Figure 3—figure supplement 1*]). The estimated counts of these transmission events are consistent with the estimated inter-species transition rates and demonstrate that within-species transmission occurs at a higher rate. Specifically, badger-to-badger transmission was estimated to occur at least 2.7 times more frequently than badger-to-cattle transmission (lower 2.5%: 2.2, upper 97.5%: 3.8). In cattle, analyses estimated that at least 46 cattle-to-cattle transmission events occurred (lower 2.5%: 40, upper 97.5%: 56), whereas the estimated number of cattle-to-badger events bounded zero (lower 2.5%: 0, upper 97.5%: 4, with a median value of zero). The counts of events between individual animals outputted by BASTA represent the lower bound of the number of transmission events that occurred over the evolutionary

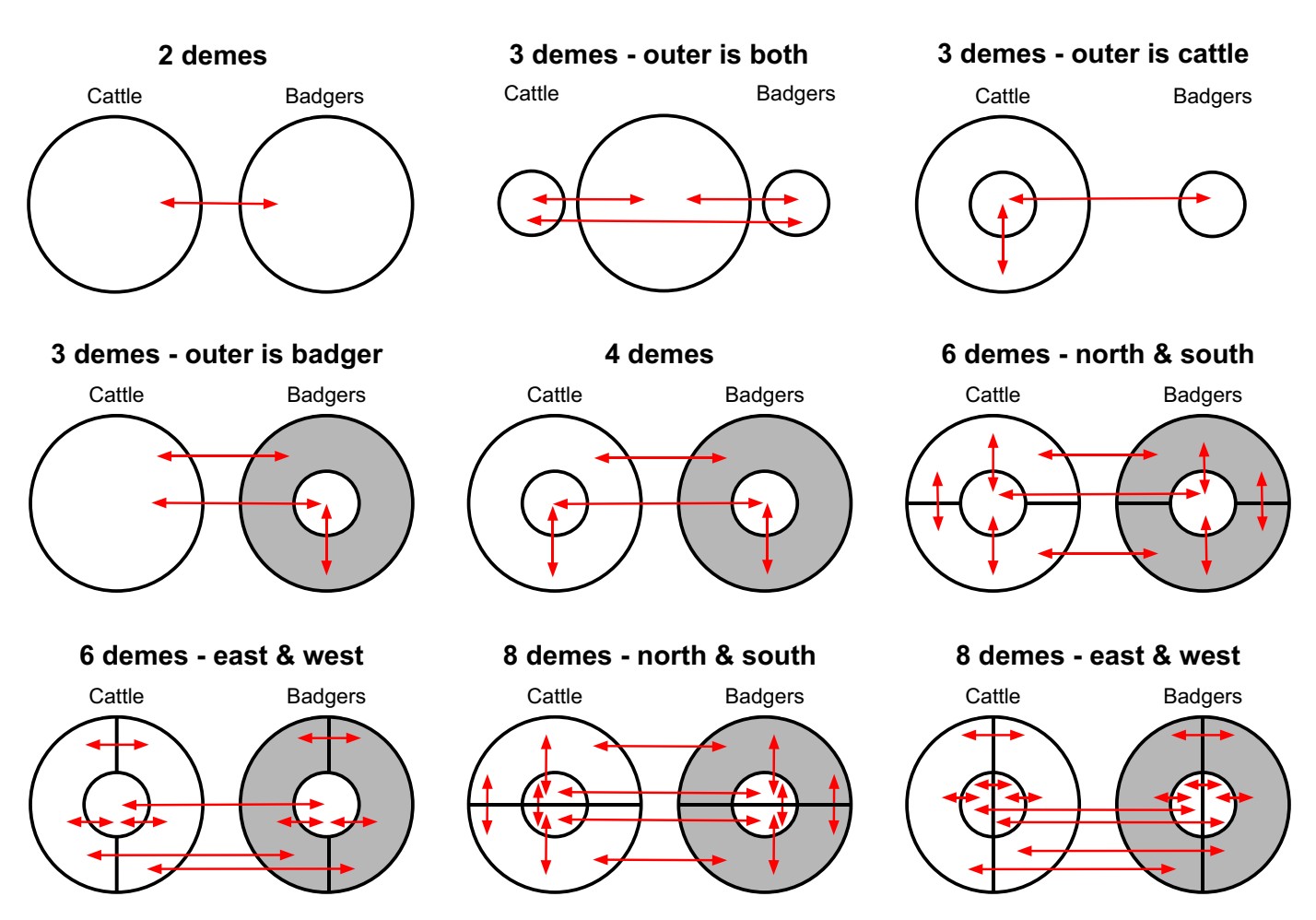

**Figure 6.** Deme assignment diagrams illustrating the different demes (sub-populations) defined in a range of structured population analyses conducted using BASTA. In each analysis, the *Mycobacterium bovis* sequences available were assigned to each deme based upon the sampled species and their sampling location. The grey doughnut in the badger demes represents an un-sampled population. These diagrams are based on the spatial associations of the badger and cattle-derived *M. bovis* sequences shown in *Figure 5*.

history of the sampled *M. bovis* population because they are estimated on the transmission chains between the sampled and ancestral host animals and do not account for missing individuals in these chains.

Taken together, the results from the BASTA analyses are consistent with the hypothesis that circulation of *M. bovis* in our study populations involved transmission within and between the badgers and cattle. In addition, the directional inter-species transmission rates indicate that transmission from badgers to cattle occurred more frequently than transmission from cattle to badgers and inter-species transmission rates were estimated to be considerably lower than intra-species transmission rates.

## Discussion

We hypothesised that the sampled *M. bovis* population was circulating within and between the sampled cattle and badger populations. Testing our hypothesis across multiple analyses, we found that, while none of these analyses are definitive in their own right, our results are consistent with our hypothesis and suggest that there has been a long history of within- and between-species transmission in the Woodchester Park area, and an important role for badgers in disease persistence.

Our choice of analytical methods was based in part on our awareness of underlying data biases. Ideally, sampling should be proportionate to prevalence in the host populations and matched over the same spatial and temporal ranges. Here, the combination of poor sensitivities of the standard tests for cattle (~50–80%; *de la Rua-Domenech et al., 2006*) and badgers (~50–70%; *Chambers et al., 2009*) and a reliance on historical archived isolates, meant data biases were unavoidable. Counterbalancing this weakness are the dense sampling of both host populations and the exceptionally detailed metadata.

Random Forest and Boosted Regression models identified strong epidemiological signatures of *M. bovis* transmission within and between host populations. Within species, metrics capturing the spatial, temporal, and network dynamics were all highly informative, indicative of *M. bovis* circulation being dependent on these factors. Between species, the variation observed between *M. bovis* sourced from cattle and badgers was found to be well explained by where the animals resided and when they were infected. Changes in these relationships could be exploited to rapidly identify changes in the epidemiology, as might be caused by badger social perturbation under culling operations (*Tuyttens et al., 2000*; *Woodroffe et al., 2006*).

The present study identified further evidence of within- and between-species transmission in the phylogenetic relationships between the *M. bovis* genomes (*Figure 1*). Five clades containing highly similar *M. bovis* genomes derived from infected cattle and badgers were identified, suggesting that substantial inter-species transmission had occurred. The presence of clades dominated by a single host species was also consistent with sustained within-species transmission. However, these phylogenetic relationships are particularly sensitive to sampling biases and should be interpreted with caution. For example, one interpretation of the basal location of the cattle-derived *M. bovis* genomes in the clades shown in *Figure 1* is that they originated in cattle. Alternatively, this pattern could be the result of sampling the cattle population over a broader temporal range (from 1988 to 2013) than the badgers (2000 to 2011).

Further interrogation of the cattle and badger life histories associated with clade 4 (*Figure 1*) revealed evidence of prolonged persistence of this lineage in the badger population (*Figure 2*). Despite the cattle population being sampled over a longer time period, the badgers associated with clade 4 were predominantly infected earlier than the cattle and that strain persisted in the badgers for over 10 years. The remaining clades examined suggested that cattle could have been infected before badgers; however, it was not possible to determine whether badgers outside of Woodchester Park could be driving these interactions. Our results do suggest that inter-badger transmission is likely to be dominated by short-range interactions, given that short spatial distances (all less than 3.7 km) were highly informative in describing the genetic relationships examined in the machine learning analyses. Therefore, badgers further away from Woodchester Park are unlikely to be directly driving the patterns observed in our sampled badger population, and the 'invading' clades observed here are more parsimoniously explained by introductions of *M. bovis* from cattle. An additional limitation of these analyses is that no other wildlife species were sampled. Previous research by *Delahay et al. (2007)* found other mammal species infected with *M. bovis* in the area, albeit at lower prevalence (7.2% in Fallow deer and 6.8% in Muntjac deer) than the sampled badger population (~30%; *Delahay et al., 2013*).

Given considerable evidence in the present study for inter-species transmission of *M. bovis*, we next used BASTA, an analysis platform that can account for sampling biases (*De Maio et al., 2018*), to quantify these processes (*Figure 3b*). The BASTA analyses estimated transition rates between demes within a structured population. As the demes within the structured model were species-specific, the estimated between-species transition rates can be considered equivalent to transmission rates between populations of badgers and cattle. The most favoured two-deme model estimated badgers-to-cattle transmission rates were, on average, 10.4 times higher than cattle-to-badgers transmission rates (*Figure 3a and b*). However, the second most favoured four-deme model (which included a more complex population structure) estimated that inter-species transmission rates were close to equal. Although even structured coalescent models do not accurately reflect spatial contact patterns, that the simplest 'two-deme' model is favoured is encouraging (i.e. more spatially structured models do not perform better). However, the two-deme model may also have been favoured because of the limited genetic diversity available to estimate the evolutionary parameters and therefore further exploration with explicitly spatial approaches is an important next step.

In the process of quantifying inter-species transmission rates, the BASTA analyses also provide counts of the number of transmission events within and between the sampled badgers and cattle (*Figure 3c*). These counts provide a conservative estimate of the minimum number of transitions between the sampled animals and their ancestors. Although it is not appropriate to directly compare the counts within- and between-species, they do demonstrate that, at a minimum, within-species transmission occurs at least twice as frequently as between-species transmission. The high degree of within-species transmission estimated here is consistent both with the results of other studies that highlight the importance of cattle-to-cattle transmission (*Costello et al., 1998*; *Gilbert et al., 2005*; *Goodchild and Clifton-Hadley, 2001*; *Green et al., 2008*; *Menzies and Neill, 2000*), and the persistent long-term infection observed in the Woodchester Park badger population (*Delahay et al., 2013*).

The high-density badger population in Woodchester Park is likely to be similar to populations found in other parts of southwest England (*Judge et al., 2017*). However, broader representativeness should be confirmed by comparison to sympatric cattle and badger populations elsewhere in Britain and Ireland, particularly in areas with high bTB incidence. In addition, we selected only isolates of spoligotype SB0263, as this was the dominant type in the badger population. The selection of SB0263 could artificially inflate the badgers-to-cattle transition rates estimated here, as the high prevalence of this spoligotype in the badgers could be a reflection of host preference. However, though there are known phenotypic differences between spoligotypes, there is no evidence that these fundamentally change the epidemiology (*Garbaccio et al., 2014*; *Wright et al., 2013*). In addition, many different *M. bovis* spoligotypes have been observed in sympatric badger and cattle populations (*Smith et al., 2003*) and SB0263 is not only one of the most common spoligotypes in the UK (*Smith et al., 2003*), it is also highly prevalent in the cattle around Woodchester Park.

If the transmission interactions estimated in our research are replicated elsewhere, this could help to explain the failure of efforts to address recurrent and persistent infection in cattle herds that coexist with badger populations (*Gallagher et al., 2013*; *Karolemeas et al., 2011*). In addition, the bidirectional transmission of *M. bovis* between species has the potential to combine local persistence in badgers with the long-distance mobility of the cattle. In line with a recent evidence review (*Godfray et al., 2018*), our research also suggests that coordinated bTB control in both cattle and badgers may be necessary to control infection in cattle. More generally, our analyses illustrate the complex interplay that underpins multi-host pathogen problems and demonstrate that, despite this complexity, appropriately defined suites of methods can be used to overcome issues of data biases and identify important epidemiological properties of these systems.

## Materials and methods

### Analyses layout

*Figure 4* describes the complete set of analyses conducted on the *M. bovis* whole genome sequences sourced from infected cattle and badgers living in and around Woodchester Park. These analyses are described in the sections that follow.

### Selecting the isolates

Since 1976, the Woodchester Park badger population has been the subject of a capture-mark-recapture study whereby each badger social group is trapped four times a year (*Delahay et al., 2013*). Social group territories are delineated annually using bait-marking (*Delahay et al., 2000*). During trapping operations, each captured badger is given a unique tattoo and at each capture event a number of samples are obtained to determine *M. bovis* infection status (full details described in *Delahay et al., 2013*). From 1990 onwards, any *M. bovis* isolated from samples taken during trapping were spoligotyped (spacer-oligo typing) using conventional methods (*Aranaz et al., 1996*) and archived. Spoligotyping reports the presence or absence of 43 known spacer sequences within a single direct repeat region of the *M. bovis* genome. In total, 230 isolates were available from the archive, which originated from samples taken from 116 different badgers from 2000 to 2011.

The cattle herds surrounding Woodchester Park undergo statutory annual testing for *M. bovis* infection as a part of routine surveillance, and results are stored in APHA's cattle testing (SAM) database (*Lawes et al., 2016*). Test-positive cattle are slaughtered, selected tissues taken for culture and

any *M. bovis* isolates are spoligotyped and archived. In addition, the movements of every cow in the UK are recorded in the Cattle Tracing System (CTS). For the present study 124 cattle-derived *M. bovis* isolates, each collected from an individual cow between 1988 and 2013, were selected from the archives. Cattle isolates were selected if they were of the same spoligotype as the badger isolates and were from herds within 10 km of Woodchester Park. More than 90% of the badger-derived isolates were spoligotype SB0263. More than 75% (1096/1442) of the isolates available from cattle within 10 km of Woodchester Park shared the same spoligotype and it is the second most common type found across England (*Smith et al., 2003*; *Smith et al., 2006*). To increase the chances of sequencing strains that were shared with the badgers in Woodchester Park, rather than circulating in the cattle population independently, only cattle-derived isolates of spoligotype SB0263 were selected. Additional spoligotype SB0263 isolates from cattle that lived in herds within 100 km of Woodchester Park (n = 65) were included to provide a broader spatio-temporal context, resulting in a total of 189 isolates.

## Generating and processing the sequencing data

Badger-derived *M. bovis* isolates were prepared for sequencing by the Agri-Food and Biosciences Institute in Northern Ireland (AFBI-NI) and for the cattle-derived isolates by APHA. *M. bovis* isolates were selected from the frozen archives and re-cultured on Löwenstein-Jensen medium. Prior to DNA extraction the isolates were heat killed in a water bath at 80°C for a minimum of 30 min. DNA was extracted from these cultures using standard high salt and cationic detergent cetyl hexadeycl tri-methyl ammonium bromide (CTAB) and solvent extraction protocols (*Parish and Stoker, 2001*; *van Soolingen et al., 2001*). Extracted DNA was sequenced at the Glasgow Polyomics facility using an Illumina Miseq producing 2 × 300 bp paired end reads (badger derived isolates) and at the APHA central sequencing unit in Weybridge using an Illumina Miseq producing 2 × 150 bp paired end reads (cattle derived isolates). The 65 additional cattle-derived isolates were sequenced at the APHA central sequencing unit in Weybridge using an Illumina NextSeq producing 2 × 150 bp paired end reads (cattle-derived isolates).

Following quality assessments in FASTQC (v0.11.2; *Andrews, 2010*; RRID:SCR_014583), the raw WGS data were trimmed using PRINSEQ (v0.20.4; *Schmieder and Edwards, 2011*; RRID:SCR_005454) and adapters were removed using TRIMGALORE (v0.4.1; *Krueger, 2015*; RRID:SCR_016946). The trimmed data were aligned to the *M. bovis* reference genome (AF2122/97; *Malone et al., 2017*) using the Burrows-Wheeler aligner (BWA, v0.7.17; *Li and Durbin, 2009*; RRID:SCR_010910). Regions encoding proline-glutamate and proline-proline-glutamate surface proteins, or annotated repeat regions were excluded (*Sampson, 2011*). Mapping quality information on all the SNVs identified was retained for each isolate.

The allele frequencies at each position in the aligned (against reference) sequence from each isolate were examined. For a haploid organism these frequencies are expected to be either 0 or 1, with some random variation expected from sequencing errors (*Sobkowiak et al., 2018*). A heterozygous site was defined as one where the allele frequencies were >0.05 and <0.95. Four cattle-derived sequences that had more than 150 heterozygous sites, and allele frequencies that were clustered and non-random (data not shown), were removed. In addition, 26 badger-derived and 16 cattle-derived *M. bovis* sequences were removed because of suspected errors in the metadata (Appendix 1: Investigating isolate metadata discrepancies).

For the sequences from the remaining isolates (204 badger- and 169 cattle-derived isolates), alleles were called at each variant position if they had mapping quality $\geq$30, high-quality base depth $\geq$4 (applied to reverse and forward reads separately), read depth $\geq$30, and allele support $\geq$0.95. For any site that failed these criteria, if the allele called had been observed in a different isolate that had passed, a second round of filtering was conducted using a high-quality base depth of 5 (total across forward and reverse reads) and the same allele support. As recombination is thought to be extremely rare for mycobacteria (*Namouchi et al., 2012*), variants in close proximity could indicate a region that is difficult to sequence or under high selection. To avoid calling variants in these regions, variant positions within 10 bp of one another were removed. Following filtering, sequences from 11 badger and 10 cattle isolates that had insufficient coverage (<95%) of the variant positions were removed. Once the alignment was generated, sites with a consistency index less than 1, generally considered homoplasies (*Farris, 1989*), were removed (n = 4, of 14,991 sites) using

*HomoplasyFinder* (v0.0.0.9; *Crispell et al., 2019*; RRID: SCR_017300). All the scripts necessary for the processing of the WGS data are freely available online.

## Comparing genetic and epidemiological distances

Our research hypothesized that within- and between-species transmission was occurring in the study system. If bi-directional transmission was occurring, then there should be epidemiological signatures in the genomic data linked to these events. These signatures are likely to relate to the spatial, temporal, and network dynamics of the sampled badger and cattle populations, as these will determine their contact patterns.

To investigate whether there were any epidemiological signatures of within- and between-species transmission of the sampled *M. bovis* isolates, the genetic distances between sequences were compared to epidemiological metrics describing the spatial, temporal, and network relationships between the animals associated with each sequence. Inter-sequence genetic distances were calculated, for every pair of sequences, by dividing the number of differences present between the pair of sequences by the total number of sites considered (n = 14,987). In addition, epidemiological metrics were calculated to identify any similarities among animals associated with a particular pair of isolates. Epidemiological metrics were calculated using the data, where available, on each animal obtained from its capture or movement and testing history (further details in Appendix 1: Defining the epidemiological metrics). Two additional dummy metrics, samples from a uniform distribution and a Boolean distribution, were included to determine a threshold of importance that distinguishes noise from signal. Inter-isolate genetic distances and associated epidemiological metrics were compared using Random Forest (RRID:SCR_015718; *Liaw and Wiener, 2002*) regression and Boosted Regression (RRID:SCR_017301; *Elith et al., 2008*) models in R (v3.4.3; *R Development Core Team, 2016*). These machine learning approaches were used to separately analyse badger–badger, badger–cattle, and cattle–cattle comparisons. For each set of comparisons, a training dataset was constructed using 50% of the data available and, following training using these data, the model was tested on the remaining 50% of the data. Genetic distances $\leq$ 15 SNVs were used for these analyses to avoid larger inter-sequence distances that were not likely to relate to the fine resolution epidemiological relationships of interest.

Random Forest and Boosted Regression approaches were selected as these methods can deal with large datasets with many highly correlated variables whose relationship to the response variable (genetic distances) cannot readily be defined (*Auret and Aldrich, 2012*). A broad range of epidemiological metrics were defined as the Random Forest and Boosted Regression models are robust to non-informative and/or highly correlated variables (*Auret and Aldrich, 2012*; *Elith et al., 2008*; *Liaw and Wiener, 2002*). The two independent approaches were used to ensure that any patterns observed were robust.

The influence of including highly correlated and non-informative predictor variables and variables with a large amount of missing data in the machine learning approaches was investigated using the Random Forest models. For highly correlated variables, clusters of correlated variables were defined and the least informative variable from each cluster was incrementally removed and the impact on the fitted Random Forest regression models was examined. A similar approach was used twice more to evaluate the influence of retaining non-informative predictor variables and of including predictor variables with large amounts of missing data in the models.

## Building phylogeny and interrogating clusters

Following investigation of population level epidemiological signatures in the sequence data, a phylogenetic tree was constructed to describe the evolutionary relationships among our set of *M. bovis* genome sequences. If inter- and intra-species transmission events were occurring in the sampled system, there should be evolutionary signatures in the phylogenetic tree. For example, if *M. bovis* sequences sourced from cattle and badgers have a very close phylogenetic relationship, this suggests that inter-species transmission has occurred. The phylogeny was constructed with the maximum likelihood algorithm in RAxML (v8.2.11; *Stamatakis, 2014*; RRID:SCR_006086) using a GTR (generalized time reversible) substitution model with 100 bootstraps. The maximum likelihood algorithm was selected as a fast alternative to Bayesian approaches. Although Bayesian approaches will better explore the phylogenetic tree space, this space is expected to be small for phylogenies based

on *M. bovis* data given its highly conserved genome. The GTR model was the most appropriate based on analyses using the `modelTest()` function in the R package PHANGORN (v2.3.1; *Schliep, 2011*; RRID:SCR_017302).

Based on the range of SNV thresholds (3–12) used to define recent *M. tuberculosis* transmission (*Bryant et al., 2013*; *Jajou et al., 2018*; *Roetzer et al., 2013*; *Yang et al., 2017*), clades containing highly related (<10 SNVs apart) cattle-derived and badger-derived sequences (inter-species clades) were identified (*Figure 1*). The testing histories and recorded movements (for cattle), and capture information (for badgers) of the sampled and in-contact animals associated with each cluster were available. These data were investigated to determine whether they provided any additional evidence to support the phylogenetic relationships indicative of inter-species transmission. 'In-contact' animals were defined as those badgers that resided in the same badger social group, or those cattle that lived in the same herd, at the same time as one or more of the sampled badgers or cattle (respectively) associated with a particular inter-species clade.

## Estimating inter-species transmission rates

To further investigate patterns of inter- and intra-species transmission, additional evolutionary analyses were completed to estimate directional inter-species transmission rates and quantify their frequency relative to intra-species transmission events. A subset of the sequences available (from 97 badger- and 83 cattle-derived isolates) was selected to estimate the transmission rate of *M. bovis* between the sampled cattle and badger populations. The selected sequences were within the parent clade containing all the inter-species clades (shown in *Figure 1*) and were sampled from within 10 km of Woodchester Park between 1999 and 2014. The subset of sequences was split into 'inner' and 'outer' groups, based on a 3.5 km radius from Woodchester Park (*Figure 5*). The 3.5 km radius size was selected to contain the sampling locations associated with all the badger-derived sequences and the closest cattle-derived sequences, based on the reported home-ranges of badgers in southern England being <1 km$^2$ (*Garnett et al., 2005*; *Macdonald et al., 2008*; *Roper et al., 2003*).

The presence of a temporal signal among the selected *M. bovis* sequences was examined (Appendix 2: Testing the presence of a temporal signal). A temporal signal was supported by a positive trend, calculated within TEMPEST (v1.5; *Rambaut et al., 2016*; RRID:SCR_017304), between each sequence's root-to-tip distance and its sampling time and the results of a tip-date randomisation procedure (*Firth et al., 2010*).

The Bayesian Structured coalescent Approximation (BASTA v2.3.1; *De Maio et al., 2015*; RRID: SCR_017303) tool, available in BEAST2 (Bayesian Evolutionary Analysis by Sampling Trees – v2.4.4 (*Bouckaert et al., 2014*), RRID:SCR_017307), uses an approximation of the structured coalescent approach (*Vaughan et al., 2014*) to estimate migration rates within a structured population. The structured population in the current context is the *M. bovis* population, whose structure was likely to relate to host species and their spatial relationships. BASTA, in contrast to previously popular methods such as discrete trait analyses (*Lemey et al., 2009*; *Pagel et al., 2004*), can estimate the ancestral structure of the population in the presence of biased sampling (*De Maio et al., 2015*). There were two biases associated with the set of sequences available. First, the prevalence of *M. bovis* in the sampled cattle and badger populations was likely to be different as a result of the on-going control operations in the cattle, therefore the sampling proportions of these different populations relative to the prevalence of *M. bovis* were likely to be unequal. Second, although the badger population within Woodchester Park has been intensively monitored and sampled, the surrounding badger population is less well understood and unsampled, whereas cattle both within and outside the Woodchester Park area have been sampled.

Based on the 'inner' and 'outer' populations of the sampled cattle and badgers (shown in *Figure 5*), a series of BASTA analyses, splitting the sampled *M. bovis* population into different demes, were designed to estimate the inter-species transition rates while accounting for the two sampling biases discussed (*Figure 6*). For each of the nine separate population structures, two separate analyses were conducted, one where the deme sizes were constrained to be equal and another where they were allowed to vary. Each of these 18 analyses was repeated three times and estimates were combined across replicates. The inter-species transition rates from each model were compared using the Akaike's Information Criterion through Markov Chain Monte Carlo (AICM; *Baele et al., 2013*), for further details see Appendix 2: Structured coalescent analyses using BASTA.

## Code availability

All the code generated for this manuscript is freely available on GitHub. General scripts are available within the 'WoodchesterPark' of the GeneralTools repository (https://github.com/JosephCrispell/GeneralTools; *Crispell, 2019a*; copy archived at https://github.com/elifesciences-publications/GeneralTools). The Java source code files can be found in a separate respository (https://github.com/JosephCrispell/Java; *Crispell, 2019b*; copy archived at https://github.com/elifesciences-publications/Java). These scripts are licenced under the General Public Licence v3.0.

## Data availability

All WGS data used for these analyses have been uploaded to the National Centre for Biotechnology Information Short Read Archive (NCBI-SRA: PRJNA523164). Because of the sensitivity of the associated metadata, only the sampling date and species will be provided with these sequences.

## Acknowledgements

We thank APHA and DEFRA staff for providing valuable comments on the manuscript, particularly James McCormack and Eleftheria Palkopoulou. Thanks to Stephen Gordon for providing access to a high-performance computing cluster maintained at the University College Dublin and advice during the generation of the manuscript. Glyn Hewinson holds a Ser Cymru II Research Chair. Daniel J Wilson is a Sir Henry Dale Fellow, jointly funded by the Wellcome Trust and Royal Society. Samantha Lycett was supported by BBSRC Institute Strategic Programme Grant to the Roslin Institute, University of Edinburgh: Control of Infectious Diseases and by a University of Edinburgh Chancellor's Fellowship. Joseph Crispell conducted the analyses as part of his BBSRC funded PhD studentship and post-doctoral position at the University of Glasgow and continued the research at his current position as a Post-Doctoral Research Fellow at University College Dublin funded on an SFI grant. Daniel Balaz is funded by the BBSRC as a post-doctoral researcher. Rowland Kao was initially supported by a Wellcome Trust Senior Research Fellowship. The long-term study of badgers at Woodchester Park is funded by DEFRA. Thanks to the reviewers and editors for advising on manuscript revisions.

## Additional information

### Funding

| Funder | Grant reference number | Author |
|---|---|---|
| Science Foundation Ireland | 16/BBSRC/3390 | Joseph Crispell |
| Wellcome Trust | 714 101237/Z/13/Z | Daniel J Wilson |
| Biotechnology and Biological Sciences Research Council | BBS/E/D/20002173 | Samantha J Lycett |
| Biotechnology and Biological Sciences Research Council | BB/L010569/1 | Joseph Crispell<br>Daniel Balaz<br>Rowland Raymond Kao |
| Biotechnology and Biological Sciences Research Council | BB/L010569/2 | Daniel Balaz<br>Rowland Raymond Kao |
| Wellcome Trust | 081696/Z/06/Z | Rowland Raymond Kao |
| Biotechnology and Biological Sciences Research Council | BB/P010598/1 | Samantha J Lycett<br>Rowland Raymond Kao |

The funders had no role in study design, data collection and interpretation, or the decision to submit the work for publication.

### Author contributions

Joseph Crispell, Formal analysis, Visualization, Methodology, Writing—original draft, Writing—review and editing; Clare H Benton, Hannah Trewby, Data curation, Methodology, Writing—review and editing; Daniel Balaz, Roman Biek, Samantha J Lycett, Methodology, Writing—review and editing; Nicola De Maio, Daniel J Wilson, Software, Methodology, Writing—review and editing; Assel

Ahkmetova, Adrian Allen, Eleanor L Presho, James Dale, Glyn Hewinson, Javier Nunez-Garcia, Robin A Skuce, Data curation, Writing—review and editing; Ruth N Zadoks, Supervision, Methodology, Writing—review and editing; Richard J Delahay, Conceptualization, Data curation, Methodology, Writing—review and editing; Rowland Raymond Kao, Conceptualization, Data curation, Supervision, Funding acquisition, Methodology, Writing—review and editing

### Author ORCIDs

Joseph Crispell (iD) https://orcid.org/0000-0002-0364-7112
Daniel Balaz (iD) https://orcid.org/0000-0003-3958-8748
Samantha J Lycett (iD) http://orcid.org/0000-0003-3159-596X
Daniel J Wilson (iD) http://orcid.org/0000-0002-0940-3311
Ruth N Zadoks (iD) http://orcid.org/0000-0002-1164-8000
Rowland Raymond Kao (iD) https://orcid.org/0000-0003-0919-6401

### Decision letter and Author response

Decision letter https://doi.org/10.7554/eLife.45833.sa1
Author response https://doi.org/10.7554/eLife.45833.sa2

## Additional files

### Supplementary files

• Transparent reporting form

### Data availability

All sequence data used for these analyses has been uploaded on to the National Centre for Biotechnology Information Short Read Archive (NCBI-SRA). Due to the sensitivity of the associated meta-data, only the sampling date and species will be provided with these sequences.

The following dataset was generated:

| Author(s) | Year | Dataset title | Dataset URL | Database and Identifier |
|---|---|---|---|---|
| Crispell Joseph | 2019 | Mycobacterium tuberculosis variant bovis Raw sequence reads | https://www.ncbi.nlm.nih.gov/bioproject/PRJNA523164 | NCBI Sequence Read Archive, PRJNA523164 |

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

## Appendix 1

# Comparing genetic and epidemiological distances

## Defining the epidemiological metrics

A series of epidemiological metrics (*Appendix 1—table 1*) were designed to capture how epidemiologically related a pair of sampled animals were. These metrics were compared to the inter-sequence genetic distances using Random Forest (*Liaw and Wiener, 2002*) and Boosted Regression (*Elith et al., 2008*) models. Three different analyses were conducted based on badger–badger, cattle–cattle, and badger–cattle comparisons. The metrics were defined based on the sampling times and locations, and the life histories of the sampled animals. Life history data for the sampled badgers included all capture event data; specifically, the capture date, the social group the badger was captured in, and the bTB status were used. The bTB status of badgers at a capture event was determined using bacteriological culture (from 1982), Brock Test ELISA (from 1982), gamma-IFN test (from 2006), and Stat-Pak test (from 2006) (tests are described in *Chambers et al., 2009*). For the sampled cattle, the life histories were made up of the cattle movement data (stored in the Cattle Tracing System) and testing (stored in APHA's cattle testing SAM database) histories. As part of the national TB surveillance program, cattle are tested using the Single Intradermal Comparative Cervical Tuberculin (SICCT) test and results are stored in the SAM database. For each sampled badger, its main social group was defined as the group in which it spent the majority of its recorded life, its sampled group was that in which it was captured when it was sampled, and its infected group was that in which it was captured when infection was first detected. For each sampled cow, the main and sampled herds were identified in a similar way. Weighted adjacency matrices were constructed counting the number of cattle/badgers that moved/dispersed between each herd/social group. Using these matrices, the shortest path lengths between herds or social groups were calculated using Dijkstra's algorithm (*Dijkstra, 1959*). The epidemiological metrics for comparing each pair of sampled animals were calculated using custom Java code (available here). Random Forest and Boosted Regression models, within R, do not handle missing data explicitly, and so where it was not possible to calculate a metric because of insufficient data a '−1' was inserted. The influence of missing data was investigated by sequentially removing metrics with the highest amounts of missing data. These analyses demonstrated little effect on the variation explained by the Random Forest models or the importance rankings of the metrics (data not shown). The nature of the epidemiological metrics meant that many were highly correlated. The influence of highly correlated metrics on the Random Forest regression models was investigated by sequentially removing metrics from highly correlated clusters. These analyses demonstrated minimal effects on the importance rankings of the epidemiological metrics (data not shown).

**Appendix 1—table 1.** Epidemiological metrics capturing the **spatial**, temporal, and *network* relationships between a pair of sampled animals. Whether or not the metric was used in the badger–badger, cattle–cattle, and badger–cattle comparisons is indicated.

| Epidemiological metrics | Badger-Badger | Cattle-Cattle | Badger-Cattle |
|---|---|---|---|
| Same main [herd/social group]? | YES | YES | NO |
| Same sampled [herd/social group]? | YES | YES | NO |
| Same infected [herd/social group]? | YES | NO | NO |
| Spatial distance between main [herd/social group]s | YES | YES | YES |
| Spatial distance between sampled [herd/social group]s | YES | YES | YES |
| Spatial distance between infected [herd/social group]s | YES | NO | NO |
| Distance from closest land parcel to main [herd/social group] using centroids | NO | NO | YES |

*Appendix 1—table 1 continued on next page*

*Appendix 1—table 1 continued*

| Epidemiological metrics | Badger-Badger | Cattle-Cattle | Badger-Cattle |
|---|---|---|---|
| Distance from closest land parcel to sampled [herd/social group] using centroids | NO | NO | YES |
| Number of days overlap between the recorded lifespans | YES | YES | YES |
| Number of days overlap between the infected lifespans | YES | NO | NO |
| Number of days spent in same [herd/social group] | YES | YES | NO |
| Number of days between infection detection dates | YES | NO | YES |
| Number of days between sampling dates | YES | YES | NO |
| Number of days between breakdown dates | NO | YES | NO |
| *Number of recorded [cattle movements/dispersal events] between main [herd/social group]s* | YES | YES | NO |
| *Number of recorded [cattle movements/dispersal events] between sampled [herd/social group]s* | YES | YES | NO |
| *Number of recorded [cattle movements/dispersal events] between infected [herd/social group]s* | YES | NO | NO |
| *Shortest path length between main [herd/social group]s* | YES | YES | NO |
| *Mean number of [cattle/badgers] traversing edges of shortest path between main [herd/social group]s* | YES | YES | NO |
| *Shortest path length between sampled [herd/social group]s* | YES | YES | NO |
| *Mean number of [cattle/badgers] traversing edges of shortest path between sampled [herd/social group]s* | YES | YES | NO |
| *Shortest path length between infected [herd/social group]s* | YES | NO | NO |
| *Mean number of [cattle/badgers] traversing edges of shortest path between infected [herd/social group]s* | YES | NO | NO |
| *Number of [cattle/badgers] recorded in both main [herd/social group]s* | YES | YES | NO |
| *Number of [cattle/badgers] recorded in both sampled [herd/social group]s* | YES | YES | NO |
| *Number of [cattle/badgers] recorded in both infected [herd/social group]s* | YES | NO | NO |
| *Shortest path length between main [herd/social group]s (some [herd/social group]s excluded)* | NO | YES | NO |
| *Mean number of [cattle/badgers] traversing edges of shortest path between main [herd/social group]s (some [herd/social group]s excluded)* | NO | YES | NO |
| *Shortest path length between sampled [herd/social group]s (some [herd/social group]s excluded)* | NO | YES | NO |
| *Mean number of [cattle/badgers] traversing edges of shortest path between sampled [herd/social group]s (some [herd/social group]s excluded)* | NO | YES | NO |
| *Shortest path length between infected [herd/social group]s (some [herd/social group]s excluded)* | NO | YES | NO |
| *Mean number of [cattle/badgers] traversing edges of shortest path between main [herd/social group]s (some [herd/social group]s excluded)* | NO | YES | NO |

## Investigating isolate metadata discrepancies

The consistency between preliminary phylogenetic data and the spoligotype (spacer-oligo type) data for the cattle- and badger-derived *M. bovis* sequences was manually examined. Spoligotyping reports the presence and absence of 43 known spacer sequences within a single direct repeat region of the *M. bovis* genome. The phylogenetic relationships of 26 badger- and 16 cattle-derived sequences were inconsistent with their spoligotype data, as they were phylogenetically dissimilar to sequences sharing their nominal spoligotype. These inconsistencies were indicative of mislabelling and therefore all the cattle- and badger-

derived sequences were investigated to determine the extent and effect of any mislabelling present. As cattle- and badger-derived sequences were sourced from different archives of *M. bovis* (isolates held by APHA or the Animal and Plant Health Agency in Weybridge for cattle, and York for badgers), it was assumed that species identification was correct.

## Badger isolates

Detailed epidemiological data describing the badger population in Woodchester Park were available. Each isolate was linked to a sampled badger via a unique identifier. Epidemiological metrics were created to summarise the spatial-, temporal-, and network-based relationships between the sampled badgers. Any mislabelled sequence would be associated with the incorrect badger. Potentially mislabelled isolates were identified as those where the variation in their genetic distances to all others was poorly explained by the available epidemiological data. The inter-sequence genetic distances were compared to the epidemiological metrics using a Random Forest regression model (*Liaw and Wiener, 2002*) and a Boosted Regression model (*Elith et al., 2008*), separately in R (v3.4.3; *R Development Core Team, 2016*). Only genetic distances <15 SNVs were included in these analyses to avoid larger distances that were not likely to relate to the fine resolution epidemiological data available.

Both the Random Forest and Boosted Regression models were able to accurately predict the inter-sequence genetic distances using the epidemiological metrics (Pearson's correlation statistics of 0.8 and 0.77, respectively). The median difference between the predicted and actual inter-isolate genetic distances was calculated for each isolate. Sequences with medians in the highest 5% of values produced by the Random Forest or Boosted Regression models were identified as potentially mislabelled and removed (*Appendix 1 - table 2*). In addition, the 11 badger-derived *M. bovis* sequences whose spoligotype data did not match the phylogenetic information but were not identified by the Random Forest or Boosted Regression models were removed from any further analyses to ensure as many of the mislabelled sequences as possible were removed.

**Appendix 1—table 2.** The 15 *M. bovis* isolates whose inter-isolate genetic distances were poorly predicted (median difference between actual and predicted genetic distances outside 95% percentile) by the Random Forest and/or Boosted Regression models. Those isolates whose spoligotypes did not match the phylogenetic patterns are also listed.

| Isolate ID | Outlier - Random Forest | Outlier - Boosted Regression | Phylogenetic-Spoligotype mismatch |
| --- | --- | --- | --- |
| WB65 | YES | YES | NO |
| WB15 | YES | YES | NO |
| WB137 | NO | YES | NO |
| WB70 | YES | YES | NO |
| WB98 | YES | YES | NO |
| WB99 | YES | YES | NO |
| WB71 | NO | YES | YES |
| WB105 | YES | YES | YES |
| WB106 | YES | YES | NO |
| WB74 | YES | YES | NO |
| WB75 | YES | YES | NO |
| WB107 | NO | NO | YES |
| WB72 | NO | NO | YES |
| WB96 | YES | NO | NO |
| WB100 | YES | NO | YES |

To determine if there was evidence of further significant errors, the effect of shuffling the metadata associated with the badger-derived sequences was examined. If the extent of the mislabelling was considerable, further shuffling should not affect the accuracy of the Random Forest model. Varying proportions of the sequences were shuffled, and a Random Forest model was used to fit the resulting inter-isolate genetic distances to the epidemiological metrics. As the proportion of the isolates shuffled increased the accuracy of the Random Forest model declined rapidly (*Appendix 1—figure 1*). The accuracy of the Random Forest model was determined by estimating the proportion of the variation in the inter-isolate genetic distance distribution that was explained by the epidemiological metrics.

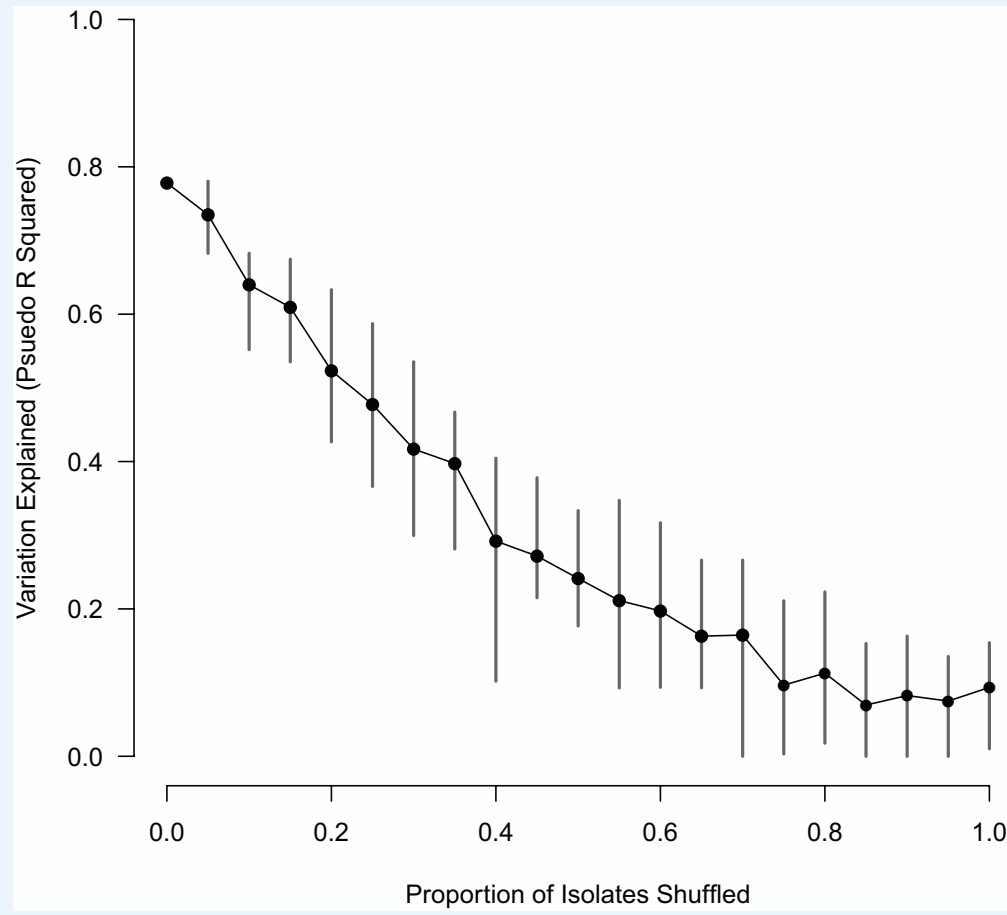

**Appendix 1—figure 1.** The impact of shuffling varying proportions of the *M. bovis* isolate sequences on the variation explained by the Random Forest model. The mean of 10 replicates is shown as a black point, with vertical lines representing the min and max values.

## Cattle isolates

The approach to investigation of the mislabelled badger-derived sequences described above was replicated for the cattle-derived sequences using modified epidemiological metrics. The Random Forest regression model (using the epidemiological metrics to explain variation in the inter-cattle-sequence genetic distances) performed well (producing an Rsq of approximately 61%) with no outlier sequences with poorly explained genetic distances being identified. As the Random Forest approach found no evidence of mislabelling in the cattle-derived *M. bovis* sequences, an additional approach was used to investigate the potential for mislabelling.

Spoligotype data were available for the cattle-derived *M. bovis* sequences. Spoligotyping indexes the presence or absence of 43 spacer sequences within a single direct repeat region

of the *M. bovis* genome (***Roring et al., 1998***). The spacer sequences present in each cattle-derived Whole Genome Sequence (WGS) were extracted and compared to a reference library of the 43 spacer sequences (***Xia et al., 2016***). The spoligotype information extracted from each *M. bovis* isolate's WGS data was compared to the spoligotype originally assigned to the isolate from conventional typing. The spoligotype information did not match that assigned from the conventional typing for 16 cattle-derived sequences. In addition, these isolates were found to be genetically more similar to sequences sharing the spoligotype derived from the WGS data than to sequences sharing the recorded spoligotype. Therefore, the spoligotype information originally available for these 16 sequences was considered incorrect – either as a result of a typing mistake or through mislabelling. These cattle-derived *M. bovis* sequences were removed. As no other spoligotype-WGS data mismatches were observed, the extent of the mislabelling in the cattle-derived sequences was considered to be minimal. In addition, as the cattle archive included isolates across a broad range of spoligotypes, there was a high chance that mislabelling would result in a different spoligotype being assigned, and the low number of wrongly assigned spoligotypes therefore indicative of a limited mislabelling.

## Metric importance in Random Forest and Boosted Regression analyses

Comparisons between the genetic distances and associated epidemiological metrics were completed using Random Forest and Boosted Regression models in R. Two different machine learning approaches were used to ensure that measures of variable importance were robust to the analyses chosen. Genetic distances $\leq$ 15 SNVs were selected to avoid between-clade comparisons and their associated large genetic distances that are unlikely to relate to fine scale epidemiological relationships. The Random Forest (and Boosted Regression) models were able to explain approximately 67% (62%), 60% (54%) and 75% (70%) of the variation in the genetic distances associated with the badger–badger (n = 12,483), cattle–cattle (n = 1927), and badger–cattle (n = 4838) comparisons, respectively. The importance of the epidemiological metrics in the Random Forest and Boosted Regression models is shown in *Appendix 1—figure 2*, *Appendix 1—figure 3*, and *Appendix 1—figure 4*. For the Random Forest models, importance was measured by the % increase in the Mean Squared Error (MSE) value when each epidemiological metric was randomly permuted. For the Boosted Regression models, the relative influence of each variable was calculated by counting the number of times each variable was selected to split the response data in a decision tree weighted by the squared improvement to the model fit that resulted from that variable being used at each split (***Elith et al., 2008***). There was good agreement between the ranking of the epidemiological metrics by the Random Forest and Boosted Regression models as both models predicted that metrics based upon the spatial, temporal, and network information were important in explaining variation in the genetic distance distribution.

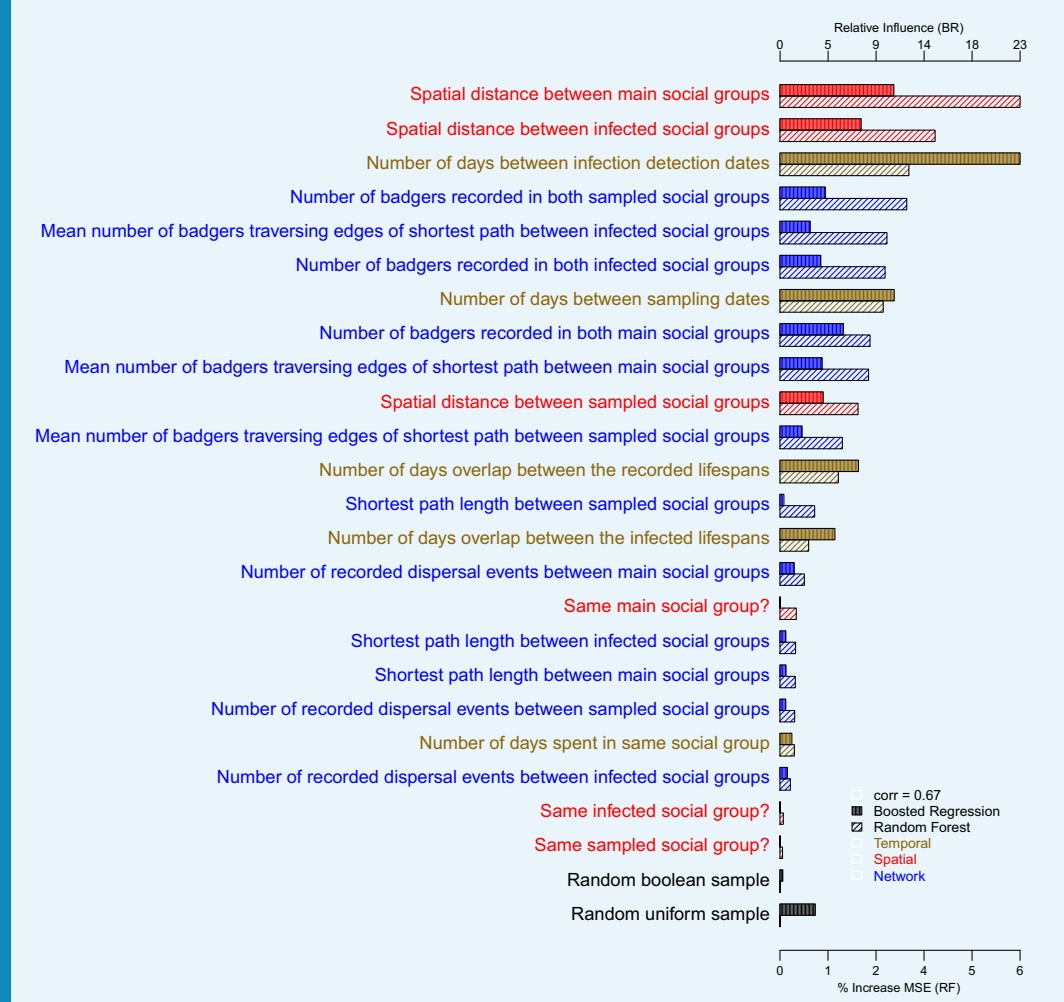

**Appendix 1—figure 2.** The importance of each epidemiological metric in explaining variation in the inter-badger-sequence genetic distance distribution. Metrics are coloured according to whether they used temporal (gold), spatial (red), or network (blue) information. The correlation (Pearson's correlation) of the variable importance from the Random Forest and Boosted Regression models is reported in the legend. Two random metrics were included, a sample from a uniform distribution and a sample from a Boolean distribution, in the regression models.

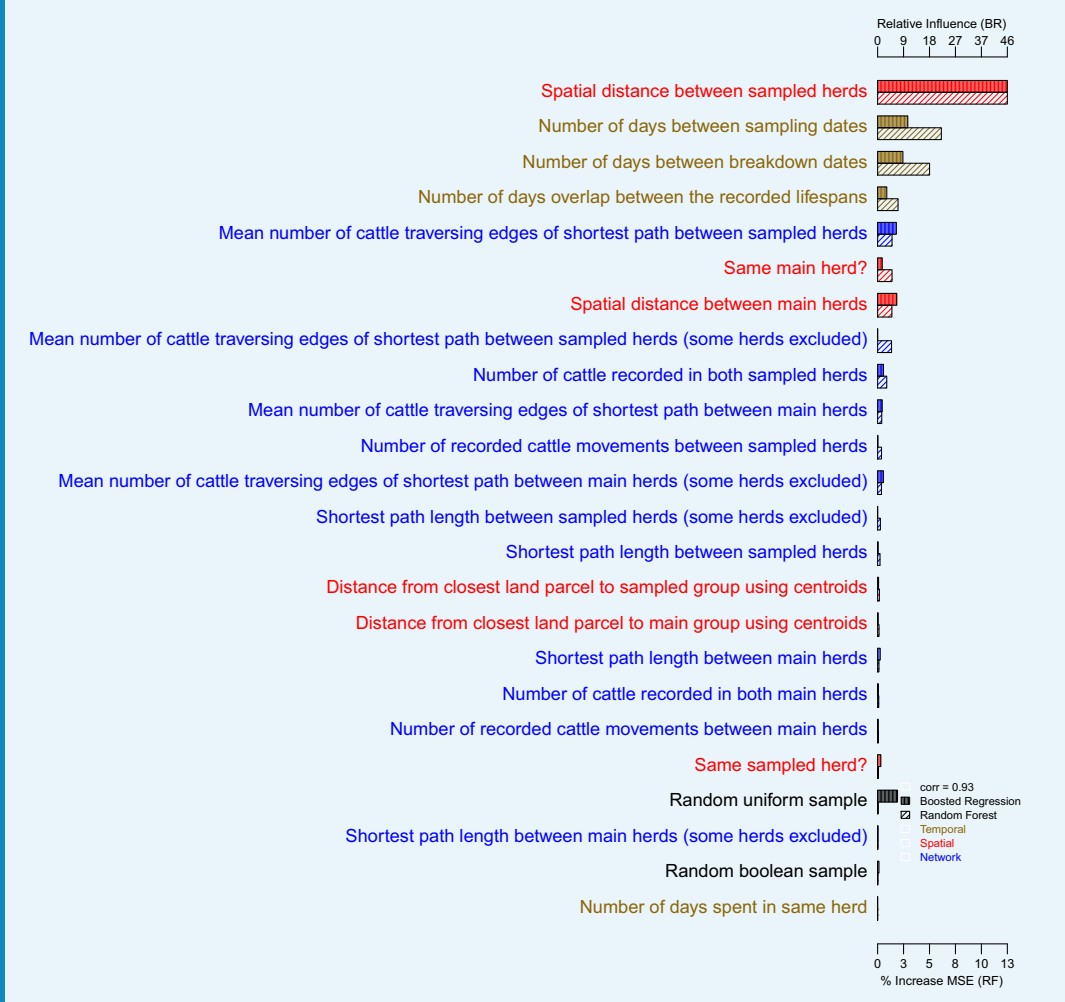

**Appendix 1—figure 3.** The importance of each epidemiological metric in explaining variation in the inter-cattle-sequence genetic distance distribution. Metrics are coloured according to whether they used temporal (gold), spatial (red), or network (blue) information. The correlation (Pearson's correlation) of the variable importance from the Random Forest and Boosted Regression models is reported in the legend. Two random metrics were included, a sample from a uniform distribution and a sample from a Boolean distribution, in the regression models.

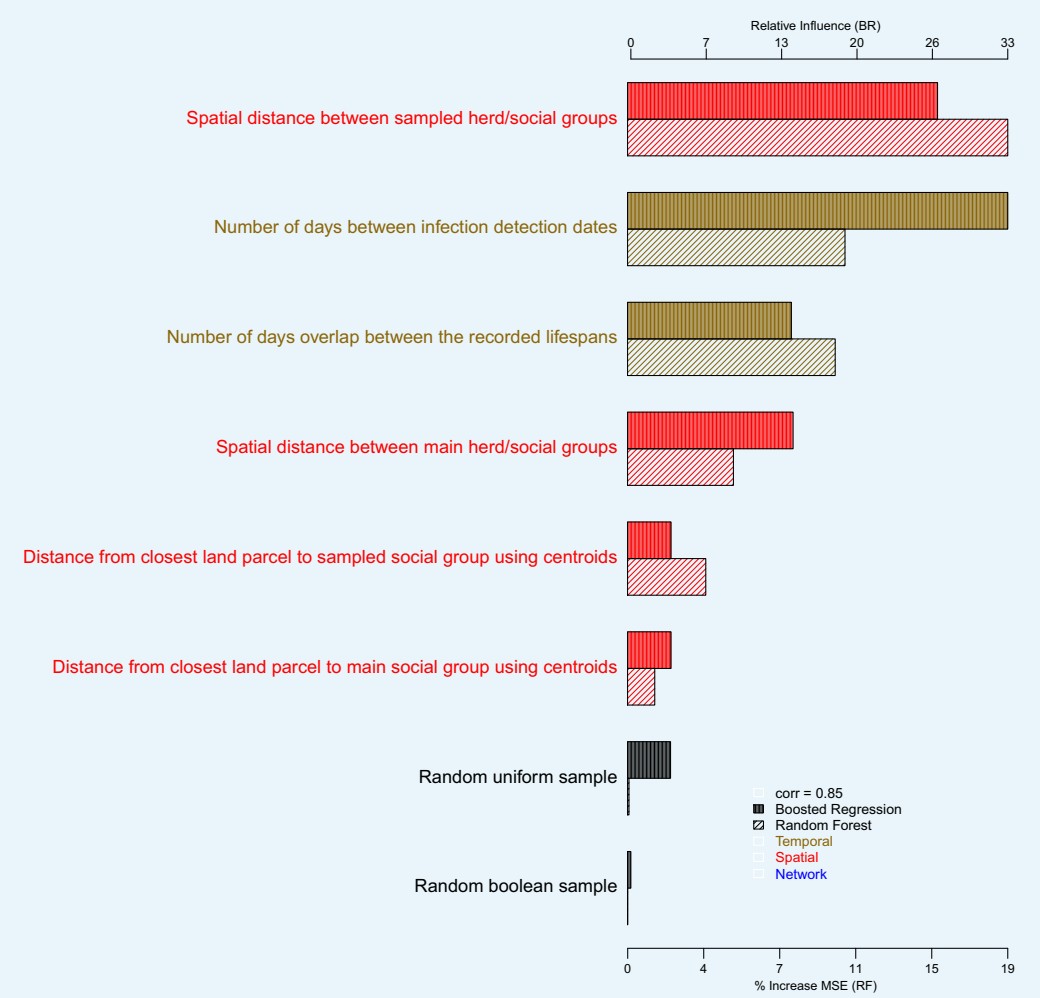

**Appendix 1—figure 4.** The importance of each epidemiological metric in explaining variation in the badger-cattle-sequence genetic distance distribution. Metrics are coloured according to whether they used temporal (gold), or spatial (red), or network (blue) information. The correlation (Pearson's correlation) of the variable importance from the Random Forest and Boosted Regression models is reported in the legend. Two random metrics were included, a sample from a uniform distribution and a sample from a Boolean distribution, in the regression models.

Partial dependence plots were used to estimate the direction of the effect between each of the epidemiological metrics (predictor variables) and the genetics distances (response variable) (*Appendix 1—figure 5*, *Appendix 1—figure 6*, and *Appendix 1—figure 7*). These relationships should be interpreted with caution as the presence of highly correlated epidemiological metrics in the data analysed will affect the accuracy of estimating the direction of effects (*Auret and Aldrich, 2012*).

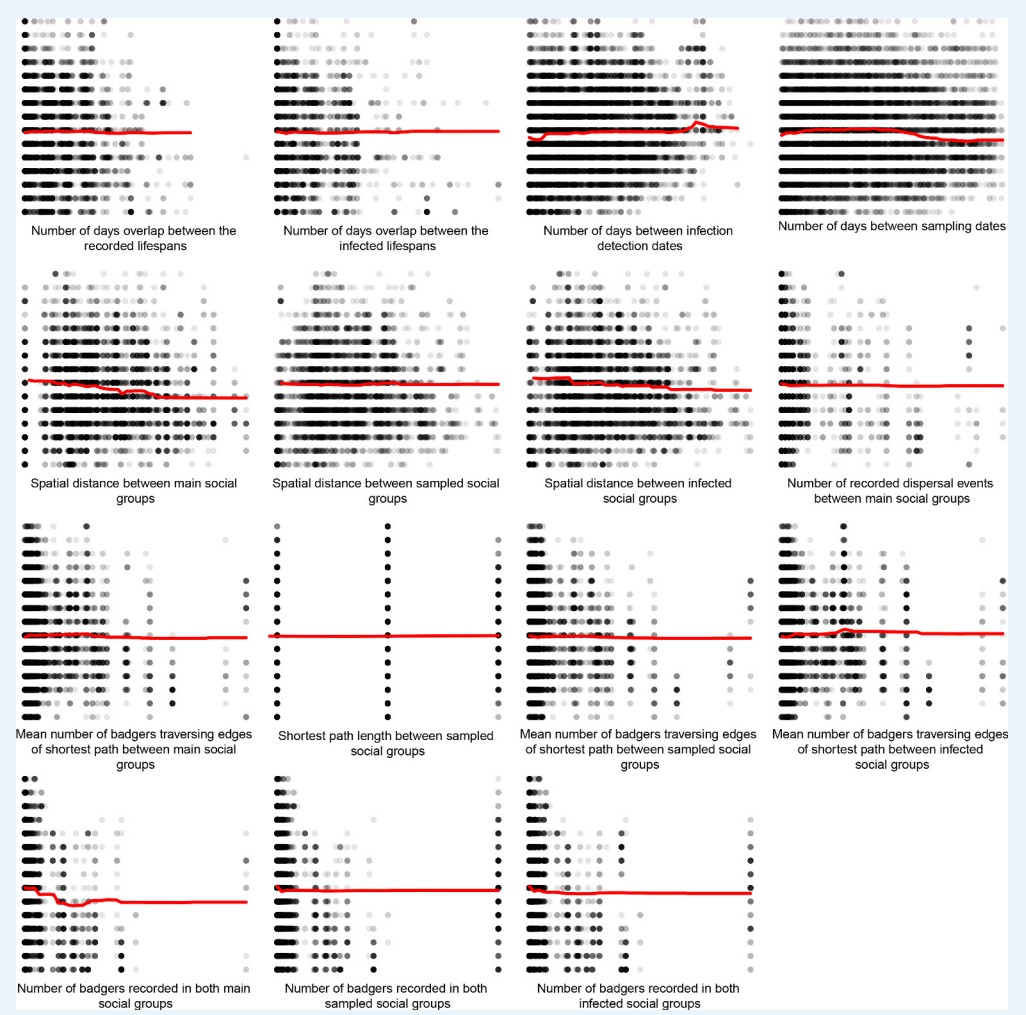

**Appendix 1—figure 5.** Partial dependence plots estimating the average marginal effect of each epidemiological metric fitted in the Random Forest regression models on the inter-badger-sequence genetic distance distribution. The Y axis in each sub-plot represents the genetic distance distribution of the number of the differences between the *M. bovis* genomes. The X axis of each plot corresponds to the range associated with the corresponding epidemiological metrics. The red line represents the average marginal effect on the predicted genetic distance for each value of the epidemiological metric. Metrics with low importance in the Random Forest models were removed (% Mean Squared Error change of < 0.5%).

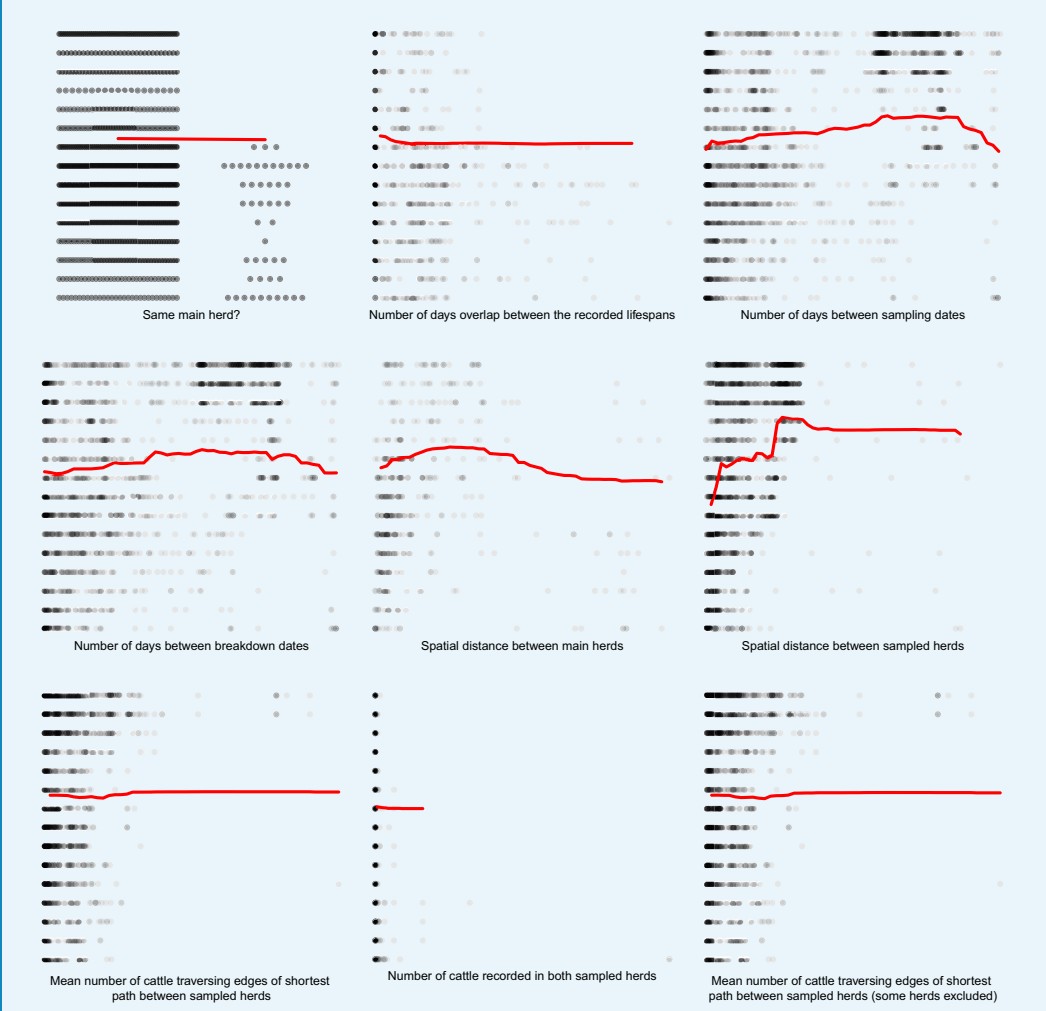

**Appendix 1—figure 6.** Partial dependence plots estimating the average marginal effect of each epidemiological metric fitted in the Random Forest regression models on the inter-cattle-sequence genetic distance distribution. The Y axis in each sub-plot represents the genetic distance distribution of the number of the differences between the *M. bovis* genomes. The X axis of each plot corresponds to the range associated with the corresponding epidemiological metrics. The red line represents the average marginal effect on the predicted genetic distance for each value of the epidemiological metric. Metrics with low importance in the Random Forest models were removed (% Mean Squared Error change of < 0.5%).

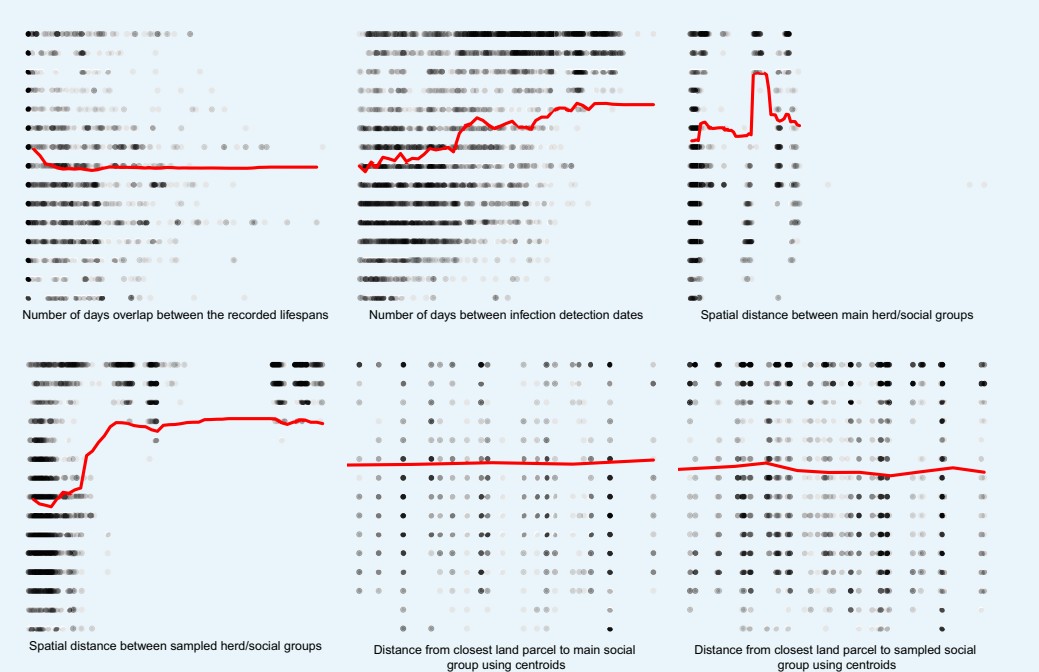

Number of days overlap between the recorded lifespans

Number of days between infection detection dates

Spatial distance between main herd/social groups

Spatial distance between sampled herd/social groups

Distance from closest land parcel to main social group using centroids

Distance from closest land parcel to sampled social group using centroids

**Appendix 1—figure 7.** Partial dependence plots estimating the marginal effect of each epidemiological metric fitted in the Random Forest regression models on the badger-cattle-sequence genetic distance distribution. The Y axis in each sub-plot represents the genetic distance distribution of the number of the differences between the *M. bovis* genomes. The X axis of each plot corresponds to the range associated with the corresponding epidemiological metrics. The red line represents the average marginal effect on the predicted genetic distance for each value of the epidemiological metric. Metrics with low importance in the Random Forest models were removed (% Mean Squared Error change of < 0.5%).

For the inter-badger distances, the partial dependence plots (*Appendix 1—figure 5*) suggest the following: as the overlap between recorded/infected lifespans increases, *M. bovis* genetic similarity decreases; as temporal distance between infection/sampling times increases, genetic similarity decreases; as the connectedness of sampled/infected/main social groups decreases, similarity decreases; and as spatial distance between infected/main/sampled social groups increases, genetic similarity increases. Although we interpret these relationships with caution, that *M. bovis* genetic similarity appears to increase as spatial distance increased is contrary to our expectation. This relationship between spatial distance and *M. bovis* similarity suggests that at the spatial scale we are examining the Woodchester Park badger population, other factors such as the connectedness of social groups could be adding noise to the spatial signal. *Vicente et al. (2007)* explored the complex relationship between social group organisation and movement and the incidence of bovine tuberculosis in the Woodchester Park badger population. *Vicente et al. (2007)* found that movement patterns within a core subset of intensively studied, and likely highly connected, social groups had a different effect to movements outside the core area, which could explain the deviations from our expectation that as spatial distance increases, *M. bovis* similarity should decrease.

For the inter-cattle distances, the partial dependence plots (*Appendix 1—figure 6*) suggest the following: cattle from the same herd were more likely to share similar *M. bovis*; more overlap in lifespans increased genetic similarity; increases in temporal and spatial distances were associated with a decrease in *M. bovis* similarity; and as network connectedness increases, *M. bovis* similarity decreases. There was considerable noise around these trends and many of the slopes were shallow and their direction of effect tended to flip with large genetic distances. One of the clearest factors influencing this noise was number of

data points, with larger genetic distances there were little data available to determine whether a discernible signal was present.

For the badger–cattle distances, the partial dependence plots (*Appendix 1—figure 7*) suggest the following: increased lifespan overlap was associated with increased *M. bovis* similarity, and as temporal and spatial distances increased, *M. bovis* similarity decreased. There was a lot of noise around these relationships, but these trends were in line with our expectations that cattle and wildlife in close proximity in time and space are more likely to transmit infection to one another.

## Appendix 2

### Phylogenetic analyses

#### Testing the presence of a temporal signal

Prior to any evolutionary analyses, which will assume a clock-like substitution rate, the existence of a temporal signal was investigated. Only those isolates selected to be analysed using BASTA (Bayesian Structured coalescent Approximation v2.3.1; *De Maio et al., 2015*) were investigated. A maximum likelihood tree was constructed and bootstrapped using RAxML (v8.2.11; *Stamatakis, 2014*). The clade containing only the selected isolates was extracted and re-rooted in TEMPEST (v1.5; *Rambaut et al., 2016*). The patristic distances (sums of the phylogenetic branch lengths) from each of the tips in the phylogeny to the root were calculated and compared to the sampling times. A slight positive trend ($R^2 = 0.12$, p value < 0.001) was observed, indicating the presence of a weak temporal signal (*Firth et al., 2010*).

The presence of a temporal signal was further examined using a tip-date randomisation procedure (*Firth et al., 2010*). A simple two-population analysis (badgers and cattle) allowing the population sizes to be different and using a relaxed clock model was used to analyse the selected isolates in BASTA. The estimated substitution rate, resulting from the BASTA analysis, was compared to those resulting from BASTA analyses where the dates associated with the isolates were randomly shuffled (*Appendix 2—figure 1*). The substitution rates estimated using randomly shuffled dates were considerably different from those estimated using the true dates, supporting the existence of a temporal signal in these data.

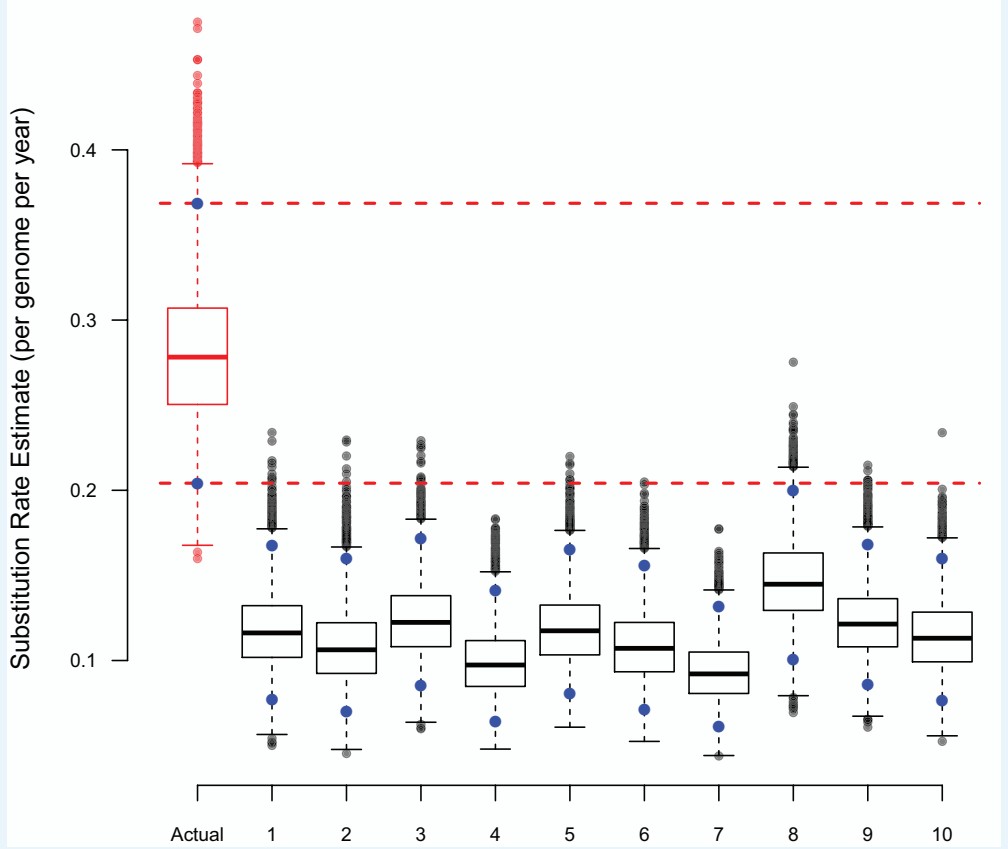

**Appendix 2—figure 1.** The substitution rate estimates from BASTA using either true or randomly shuffled sampling dates. The upper (97.5%) and lower (2.5%) bounds of each distribution

are shown as blue points, the horizontal dashed lines represent the same bounds for the estimates based on the actual dates. Each BASTA analysis using a two population (badgers and cattle) structure, allowed different but constant population sizes, and relaxed clock model based upon an HKY substitution model.

## Structure coalescent analyses using BASTA

A set of 18 BASTA analyses was derived based on different structured populations and constraining the deme (sub-populations) sizes or allowing them to vary. Each of the 18 BASTA analyses was replicated three times using a relaxed clock model and an HKY (Hasegawa-Kishino-Yano; *Hasegawa et al., 1985*) substitution model. The relaxed clock model was chosen to avoid assuming a constant evolutionary rate across the sampled *M. bovis* population and because preliminary analyses using the relaxed clock had a higher likelihood score in comparison to analyses using the strict clock (data not shown). The HKY substitution model was selected as it was the simplest such model with the lowest number of parameters that still allowed the rate of transitions and transversions to differ.

*Figure 3* in the main manuscript illustrates the inter-deme transition rate parameters estimated for each structured population, the necessity of each of these parameters was assessed using a variable rate flag parameter (either 0 or 1). If a transition rate parameter was not informative in describing the evolutionary relationships of the sampled *M. bovis* population, its flag would be frequently assigned a zero value. The estimates of inter-deme transition rates resulting from each BASTA analysis were combined across replicates, with the first 10% of the posterior samples removed. For each inter-deme transition rate posterior values were removed if their associated variable rate flag was set to zero and the remaining values were converted to forward transition rates according to *De Maio et al. (2015)*. Each BASTA analysis was repeated but the nucleotide alignment was replaced with an empty alignment. By using an empty alignment, the parameter estimates for the BASTA analysis will rely entirely on the prior distributions, this type of analysis can therefore be used to check whether the genetic data are informative for estimation of each parameter. If the estimate for a particular parameter in a BASTA analysis is similar with and without the sequence alignment, there is insufficient information in the genetic data to estimate it. For each BASTA analysis, the posterior distributions for each parameter were considerably different in the presence and absence of the sequence alignment, suggesting that there were sufficient data available to estimate them.

