## [Decision Letter]

Thank you for submitting your article "Combining genomics and epidemiology to analyse bi-directional transmission of *Mycobacterium bovis* in a multi-host system" for consideration by *eLife*. Your article has been reviewed by three peer reviewers, one of whom is a member of our Board of Reviewing Editors, and the evaluation has been overseen by Neil Ferguson as the Senior Editor. The following individuals involved in review of your submission have agreed to reveal their identity: Christian Gortazar (Reviewer #3).

The reviewers have discussed the reviews with one another, and the Reviewing Editor has drafted this decision to help you prepare a revised submission.

This is a population genomics study of the transmission of *M. bovis* between two nonhuman animal hosts, cattle and badgers, a matter of considerable ecological and practical interest.

Summary:

This is an analysis of the multispecies badger-cattle transmission system using genomic and epidemiological data to characterize the transmission of *M. bovis* between these species, a matter of considerable interest.

Strengths include the sizable and detailed data set, and the relatively clear exposition of what analyses were done.

Essential revisions:

The major required revision is to place the work in the context of clear hypotheses, predictions of these hypotheses for the results of analyses, and interpretation of the analyses as they inform our judgment on those hypotheses. As currently written, like many pathogen genomics papers, this paper presents analyses and results, but leaves the rationale for the analyses and the interpretation of the results in terms of scientific hypotheses unclear. While this is not atypical for papers in the field, it makes it very hard for an interested non-specialist in the subject matter (ecology of *M. bovis*) to appreciate the paper. For a general interest journal like *eLife*, it is problematic because the reader is left with an unclear sense of what has and has not been shown.

It seems that the major hypotheses being tested revolve around the extent to which badgers are a reservoir for cattle *M. bovis* infection. Put somewhat more precisely, finding that they are a reservoir would mean that badger-badger transmission sustains the infection, that badger-to-cattle transmission is frequent and is the source (immediate or ultimate) of most cattle infections.

There are four main types of analysis in this paper. In a revised version we would expect each of these to be motivated explicitly by "we hypothesized x and tested it using this method and found that results were/were not consistent with x”. This is the main missing item in the paper.

1) Random forest and boosted regressions. The reason for doing both is unclear, and the boosted regressions are not described much at all. The RF method seems perhaps to be testing the plausibility of the idea that genetic distance indicates likely transmission. This seems more or less borne out by the main results where spatial and social proximity have strong explanatory roles (the sign is not stated but I assume that epi distance and genetic distance are positively related). I'm not sure that I'd expect such a relationship to hold over the full scale of distances (beyond some distance I would think there would be no relationship anymore) but this is a detail. The finding that "same host" has no role in the RF is quite weird – usually isolates from the same host would be nearly identical. No explanation is given for why the many different network and spatial measures are used, or which one expects to be positive in such a complex model, especially assuming that this is multivariable, so they are all conditional on the others. Overall, the RF seems maybe to be consistent with the data being of high quality and with transmission being related to short genetic distance, but not to clearly refute or confirm any hypothesis. Please clarify why these analyses were done and what the results mean.

2) Phylogenetic reconstruction. Here most of the clades have high probability of ancestral nodes being in cattle, seemingly inconsistent with the badger-reservoir hypothesis. Please comment on how these results should be interpreted.

3) The epidemiological descriptive data, where badgers seemed to have it long before cattle. Seems consistent with the reservoir hypothesis, though badgers are also much more widely sampled. Please make this explicit (modified if it has been misunderstood).

4) The structured coalescent analyses, in which badger-to-cattle transmissions seem to be much more common than the reverse under nearly all models, including the best-supported ones. One aspect that confuses me is that presumably the different lifespans of the hosts lead to different durations of the infection, so I am not sure if number of transmissions per unit time is the best measure of transmission. But taken at face value this seems consistent with the reservoir hypothesis. Please clarify what you think the interpretation is, and in particular (two reviewers wondered) whether these can be interpreted as the ratio of rates (transmissions per unit time), of basic or effective reproductive numbers (transmissions per infection), or something else that has physical interpretation.

Assuming that the Reviewing Editor, a nonexpert in the substantive field, has understood the above correctly, please modify the discussion to give careful consideration of the contrasting observations (3 and 4 support the hypothesis, 2 argues against it) and how they can be reconciled. If the authors can convincingly do that and answer (even with uncertainty) a clear scientific question, this could become publishable.

[Editors' note: further revisions were requested prior to acceptance, as described below.]

Thank you for resubmitting your work entitled "Combining genomics and epidemiology to analyse bi-directional transmission of *Mycobacterium bovis* in a multi-host system" for further consideration at *eLife*. Your revised article has been favorably evaluated by Neil Ferguson (Senior Editor), a Reviewing Editor, and one reviewer.

This paper has been extensively revised, and the scientific logic is now far clearer. There are some remaining issues that need to be addressed before acceptance, as outlined below:

1) The sampling selected for genetically similar isolates in the two host species, which will (I believe in all cases) increase the estimated transition rates above that which is typical for all strains. For example, any spoligotypes that are not transmitted between the species will not be counted. This is an important caveat to the conclusions about the frequency of interspecies transfer and needs to be explicit in the Discussion.

2) The phylogeny and BASTA analyses document interspecies transfer in both directions. The regression trees seemingly show the relevance of within-species transmission. Neither alone nor together do they answer the question of reservoir – is transmission in either species sufficient on its own for maintenance of the infection and continuing spillover into the other? The Discussion recommends integrated control, and intuitively this seems sensible, but on their own evidence of transmission within each species and between the two does not prove that essentially R_0{ii} >1 for either species, where ii represents transmission from species i to species i, and this is a condition for i to be a reservoir. I believe that the data are formally consistent with the possibility that eradication in either species would eradicate in the other (seems unlikely, as it requires a big role for interspecies transmission) or, more plausibly, that eradication in one species would eliminate it in the other because R0 within that species <1. If this reasoning is wrong, please refute. If it is right, please note this in the discussion and soften the call for integrated control.

3) Is there any way to quantify the ratio of within to between-species transmissions? This is hinted at frequently, but the numbers are never given.

4) The inclusion of a factor in the RF and BRT analyses does not guarantee that it is included in the expected direction. Can the authors report the direction of the effect for each included factor and explain any discrepancies from expectation, e.g. that overlapping lifespan = lower distance?

5) Can the authors explain, in Figure 3 a)what "mean posterior probability of each rate" means (I think it means posterior probability that it is positive) and b) why the ratio of transition counts and ratio of transition rates is so different?

6) No clear answer was given to essential revision 4, which asked in what if any sense these transition rate ratios can be interpreted as reproductive number ratios or something else epidemiological. Please comment on this in the Discussion. Also please edit carefully to use "transition" rather than "transmission" or explain why both these terms appear (as far as I can tell interchangeably) in the text.

---

## [Author Response]

Essential revisions:The major required revision is to place the work in the context of clear hypotheses, predictions of these hypotheses for the results of analyses, and interpretation of the analyses as they inform our judgment on those hypotheses. As currently written, like many pathogen genomics papers, this paper presents analyses and results, but leaves the rationale for the analyses and the interpretation of the results in terms of scientific hypotheses unclear. While this is not atypical for papers in the field, it makes it very hard for an interested non-specialist in the subject matter (ecology of M. bovis) to appreciate the paper. For a general interest journal like eLife, it is problematic because the reader is left with an unclear sense of what has and has not been shown.It seems that the major hypotheses being tested revolve around the extent to which badgers are a reservoir for cattle M. bovis infection. Put somewhat more precisely, finding that they are a reservoir would mean that badger-badger transmission sustains the infection, that badger-to-cattle transmission is frequent and is the source (immediate or ultimate) of most cattle infections.There are four main types of analysis in this paper. In a revised version we would expect each of these to be motivated explicitly by "we hypothesized x and tested it using this method and found that results were/were not consistent with x”. This is the main missing item in the paper.

Thank you for highlighting this critical issue. The following changes were completed in response:

– Added description of our hypothesis and three objectives to the end of the Introduction.

– Added statements into the Results section to explicitly link each result to an objective and our hypothesis.

– Added similar explicit linking statements into the Materials and methods section.

1) Random forest and boosted regressions. The reason for doing both is unclear, and the boosted regressions are not described much at all. The RF method seems perhaps to be testing the plausibility of the idea that genetic distance indicates likely transmission. This seems more or less borne out by the main results where spatial and social proximity have strong explanatory roles (the sign is not stated but I assume that epi distance and genetic distance are positively related). I'm not sure that I'd expect such a relationship to hold over the full scale of distances (beyond some distance I would think there would be no relationship anymore) but this is a detail. The finding that "same host" has no role in the RF is quite weird – usually isolates from the same host would be nearly identical. No explanation is given for why the many different network and spatial measures are used, or which one expects to be positive in such a complex model, especially assuming that this is multivariable, so they are all conditional on the others. Overall, the RF seems maybe to be consistent with the data being of high quality and with transmission being related to short genetic distance, but not to clearly refute or confirm any hypothesis. Please clarify why these analyses were done and what the results mean.

Thank you for highlighting this. We have made the following changes:

– Additional analyses using Boosted Regression models were completed to allow the two methods to be more directly compared (Appendix 1—figures 2, 3 and 4 were updated and changes were made in the Results section of the main manuscript and to Appendix 1: Metric importance in Random Forest and Boosted Regression Analysis).

– Additional text was included to provide clarity regarding the use of these analyses in the Results section.

– The reference to the “same host” epidemiological metric has been removed. This metric wasn’t informative in the machine learning analyses because there were too little data available (only 201 of the 12,483 badger-to-badger comparisons were between genomes sourced from the same animal).

– The trends in relationship between each predictor variable and the genetic distances were examined using partial dependence plots. The relationships were found to be non-linear and variable and given that partial dependence plots can be misleading when highly correlated predictor variables are present in the model. Therefore, we only added broad statements about direction in the Results section.

– Additional analyses were described in the Results section, which investigated the influence of missing data and highly correlated predictor variables.

2) Phylogenetic reconstruction. Here most of the clades have high probability of ancestral nodes being in cattle, seemingly inconsistent with the badger-reservoir hypothesis. Please comment on how these results should be interpreted.

We agree that these results were inconsistent with our hypothesis. Re-examining our analyses, it was clear that the ancestral character estimation method used was highly sensitive to sampling biases and therefore these analyses were removed. The BASTA analyses we describe conducted similar analyses but BASTA is considered more robust because these analyses can account for the known sampling biases in our dataset. The following changes were made:

– Removed text referring to ancestral character estimation throughout manuscript.

– Figure 1 and its legend were updated to remove reference to the ancestral character estimation.

– Provided additional text in the Discussion noting that clades 1,2, 3, and 5, which appear to have a cattle origin, are likely to originate from outside of Woodchester Park where we have no badger isolates and therefore sampling biases may influence any observations of a cattle origin.

– Added additional explanatory text in the Results section to highlight the influence of sampling biases.

3) The epidemiological descriptive data, where badgers seemed to have it long before cattle. Seems consistent with the reservoir hypothesis, though badgers are also much more widely sampled. Please make this explicit (modified if it has been misunderstood).

The epidemiological description (Figure 2) is restricted to the animals associated with clade 4 in Figure 1. The observation that the badgers in this figure were sampled over a broader temporal window is a reflection of the clade 4 strain rather than our sampling. In fact, in our research the cattle population was sampled over a broader temporal window (1988-2013) as compared to the badger population (2000-2011). To clarify this the following changes were made:

– Additional text added to the legend of Figure 3.

– Minor changes were made to Figure 2 and its legend.

– Included four additional supplementary figures to Figure 1 (Figure 1—figure supplements 1, 2, 3, and 4) documenting the life histories of the animals associated with clades 1, 2, 3 and 5. These are referred to in the Results section.

4) The structured coalescent analyses, in which badger-to-cattle transmissions seem to be much more common than the reverse under nearly all models, including the best-supported ones. One aspect that confuses me is that presumably the different lifespans of the hosts lead to different durations of the infection, so I am not sure if number of transmissions per unit time is the best measure of transmission. But taken at face value this seems consistent with the reservoir hypothesis. Please clarify what you think the interpretation is, and in particular (two reviewers wondered) whether these can be interpreted as the ratio of rates (transmissions per unit time), of basic or effective reproductive numbers (transmissions per infection), or something else that has physical interpretation.

You are correct that the different lifespans animals will lead to durations of infection. However, the transmission rates estimated in BASTA (our structured coalescent approach) are done at the population level and for these analyses it is the time from infection to transmission that is important. In addition, the transmission rates are at the deme level rather than the individual level. Lastly, the average lifespan of dairy cattle (6.5 years) and badgers (5-8 years) are fairly similar.

In the manuscript, we had incorrectly referred to the transmission rates (Figure 3B) as “cattle-to-badger” or “badger-to-cattle”, which should have used “badgers” – this has been corrected.

Similarly, in reference to the estimation of the number of within and between species transition events (Figure 3D) we incorrectly used “badgers” and “cattle” – these have been corrected to “cow” and “badger”.

To help with the interpretation of the results of the BASTA analyses, an additional panel was added to Figure 3C. Figure 3C presents the median ratio of the badgers-to-cattle transmission rate divided by the cattle-to-badgers transmission rate estimated by each model analysed in BASTA. Additional references in the text for Figure 3C were added.

Assuming that the reviewing editor, a nonexpert in the substantive field, has understood the above correctly, please modify the Discussion to give careful consideration of the contrasting observations (3 and 4 support the hypothesis, 2 argues against it) and how they can be reconciled. If the authors can convincingly do that and answer (even with uncertainty) a clear scientific question, this could become publishable.

Thank you for this recommendation. In the revised manuscript the Discussion has been re-written. The aim of this re-write was to produce a shorter and clearer discussion that describes how each analysis and its results should be interpreted in the context of our hypothesis and the broader literature. We note that, while we understand why it seemed like the observations contradict each other, by recognising the role that biases played in the analysis 2, these are now consistent.

[Editors' note: further revisions were requested prior to acceptance, as described below.]

This paper has been extensively revised, and the scientific logic is now far clearer. There are some remaining issues that need to be addressed before acceptance, as outlined below:1) The sampling selected for genetically similar isolates in the two host species, which will (I believe in all cases) increase the estimated transition rates above that which is typical for all strains. For example, any spoligotypes that are not transmitted between the species will not be counted. This is an important caveat to the conclusions about the frequency of interspecies transfer and needs to be explicit in the Discussion.

Our analyses were limited to highly related genomes. There were very few examples of non-SB0263 isolates in the sampled badgers (>90%), and therefore, for this population, we believe our results are a good representation of the epidemiological characteristics of the system. Further, given the small number of samples of differing spoligotypes and the large genetic distances between them, we would likely be unable to improve the accuracy of our estimates even if we added these samples. However, we do agree that the selection of SB0263 may artificially inflate the importance of badger-to-cattle transmission over cattle-to-cattle transmission. This selection bias is alluded to in the Discussion:

“In addition, we have only considered spoligotype SB0263 and there are known phenotypic differences between spoligotypes, though such differences are unlikely to fundamentally change the epidemiology (Garbaccio et al., 2014; Wright et al., 2013).”

In addition, more detail is provided in the Materials and methods section:

“More than 90% of the badger-derived isolates were spoligotype SB0263. More than 75% (1096/1442) of the isolates available from cattle within 10km of Woodchester Park shared the same spoligotype and it is the second most common type found across England (Smith et al., 2003; Smith, Gordon, de la Rua-Domenech, Clifton-Hadley, and Hewinson, 2006).”

The section in the Discussion has been edited and expanded:

“In addition, we selected only isolates of spoligotype SB0263, since this was the dominant type in the badger population. […] In addition, many different *M. bovis* spoligotypes have been observed in sympatric badger and cattle populations (Smith et al., 2003) and SB0263 is not only one of the commonest spoligotypes in the UK (Smith et al., 2003), it is also highly prevalent in cattle around Woodchester Park.”

2) The phylogeny and BASTA analyses document interspecies transfer in both directions. The regression trees seemingly show the relevance of within-species transmission. Neither alone nor together do they answer the question of reservoir – is transmission in either species sufficient on its own for maintenance of the infection and continuing spillover into the other? The Discussion recommends integrated control, and intuitively this seems sensible, but on their own evidence of transmission within each species and between the two does not prove that essentially R_0{ii} >1 for either species, where ii represents transmission from species i to species i, and this is a condition for i to be a reservoir. I believe that the data are formally consistent with the possibility that eradication in either species would eradicate in the other (seems unlikely, as it requires a big role for interspecies transmission) or, more plausibly, that eradication in one species would eliminate it in the other because R0 within that species <1. If this reasoning is wrong, please refute. If it is right, please note this in the Discussion and soften the call for integrated control.

Thank you for highlighting this issue and we agree that the language used needs to be tightened.

We have replaced the use of the term ‘reservoir’ with respect to the badger population and have removed our statements suggesting the badgers are maintaining infection. Instead, we note that our evidence suggests that infection can persist in the badger population independently for over 10 years. We have softened our statement calling for coordinated control by changing “it will be necessary” to “it may be necessary” in the Discussion. Lastly, we have edited the final sentence of the abstract to state:

“If representative, our results suggest that control operations should target both cattle and badgers.”

3) Is there any way to quantify the ratio of within to between-species transmissions? This is hinted at frequently, but the numbers are never given.

It would be possible to quantify the ratio of the estimated number of within- and between-individual transmission events (from Figure 3C) but we don’t feel this is appropriate. These counts of the transmission events between individual animals can only be considered conservative estimates of the minimum number of events because they don’t account for multiple host transitions (badger or cow) on a single branch from a parent node to its child. In addition, we assumed that, where possible, the host animal represented the parent and one of the child nodes. In contrast, the more robust inter-species transmission rates are explicitly estimated at the population level and account for missed individuals on the transmission chains by allowing multiple host transitions on a single branch.

We included additional sentences describing the transmission event counts in the discussion in the Results section:

“The counts of events between individual animals outputted by BASTA represent the lower bound of the number of transmission events that occurred over the evolutionary history of the sampled *M. bovis* population because they are estimated on the transmission chains between the sampled and ancestral host animals and don’t account for missing individuals in these chains.”

Also, additional lines were added in the Discussion section:

“These counts provide a conservative estimate of the minimum number of transitions between the sampled animals and their ancestors. While it is not appropriate to directly compare the counts within- and between-species, they do demonstrate that, at a minimum, within-species transmission occurs at least twice as frequently as between-species transmission.”

Lastly, we created a supplementary figure to Figure 3 (Figure 3—figure supplement 1) that illustrates how the estimated transmission events were counted on each phylogeny in the posterior distribution of trees estimated by the two deme model in BASTA.

4) The inclusion of a factor in the RF and BRT analyses does not guarantee that it is included in the expected direction. Can the authors report the direction of the effect for each included factor and explain any discrepancies from expectation, e.g. that overlapping lifespan = lower distance?

Partial dependence plots were produced from the Random Forest analyses estimating the direction of the relationship between each epidemiological metric and genetic distances. These plots were added into Appendix 1—figures 5, 6, and 7. In addition, two new random metrics were included in the Random Forest and Boosted Regression analyses to provide an indication of the importance that could be attributed to a variable that had no relationship to the genetic distances (see updates to Appendix 1— figures 2, 3, and 4). Sentences describing the trends between the epidemiological trends and genetic distances have been included in Appendix 1:

“Partial dependence plots were used to estimate the direction of the effect between each of the epidemiological metrics (predictor variables) and the genetics distances (response variable) (Appendix 1—figure 5, Appendix 1—figure 6, and Appendix 1—figure 7). […] There was a lot of noise around these relationships, but these trends were in-line with our expectations that cattle and wildlife in close proximity in time and space are more likely to transmit infection to one another.”

5) Can the authors explain, in Figure 3 a) what "mean posterior probability of each rate" means (I think it means posterior probability that it is positive) and b) why the ratio of transition counts and ratio of transition rates is so different?

a) We apologise for the confusion. Based on the number of inter-species rates estimated, this value is either the posterior probability directly (for the first three models) or the mean calculated across the different inter-species transmission rates that were estimated and summed, as seen in Author response table 1 (derived from Figure 5).

**Author response table 1. resptable1:** 

	2 demes	3 demes – outer is both	3 demes – outer is cattle	3 demes – outer is badgers	4 demes	6 demes – north and south	6 demes – east and west	8 demes – north and south	8 demes – east and west
CB	1	1	1	2	2	3	3	4	4
BC	1	1	1	2	2	3	3	4	4

To improve clarity, text has been added to the legend of Figure 3, clarifying how the numbers representing the posterior probabilities were calculated:

“The values above each vertical line represent the posterior probability of each rate, either as a mean of probabilities associated with multiple estimated rates (for the 3Deme_outerIsBadgers, 4Deme, 6Deme and 8Deme models) or a single probability (for the 2Deme, 3Deme_outerIsBoth, and 3Deme_outerIsCattle models).”

b) As stated in our response to point 3, the transmission counts represent the lower bounds on the number of events between individual animals. In contrast, the inter-species transmission rates are estimated at the population and can’t be directly compared to these counts. In addition, because the counts don’t account for missing individuals on the transmission chains from the ancestral individuals to the sampled animals, they are susceptible to sampling biases.

To avoid confusion panel C of Figure 3 was removed. In addition, the new panel C (previously D) has been edited to improve its clarity. The counts for within species and between species events have been separated and the number of badgers and cattle sampled at the tips of the phylogenies that the counts are derived from has been noted.

6) No clear answer was given to essential revision 4, which asked in what if any sense these transition rate ratios can be interpreted as reproductive number ratios or something else epidemiological. Please comment on this in the Discussion. Also please edit carefully to use "transition" rather than "transmission" or explain why both these terms appear (as far as I can tell interchangeably) in the text.

a) The relative transmission rates are the most appropriate calculations given the evolutionary analysis methods that were used in this paper. It isn’t appropriate to discuss these in terms of the reproductive ratio because we are calculating population-level rather than individual transmission rates. We note that direct estimates of reproductive numbers are possible but would require extensive additional work using different methods.

b) In our case the inter-population transition rates estimated here can be considered inter-species transmission rates because the populations we consider are species specific. As noted above in 6a above, these are estimates at the population rather than individual animal level.

Any instances of the misuse of these terms in the manuscript were corrected. Additional text added into the discussion to clearly define our use of transmission and transition:

“The BASTA analyses estimated transition rates between demes within a structured population. Since the demes within the structured model were species-specific the estimated transition rates can be considered equivalent to transmission rates between populations of badgers and cattle.”